# Fragment-based drug nanoaggregation reveals drivers of self-assembly

Chen Chen [1,2,3], You Wu[1,2,3], Shih-Ting Wang [4], Naxhije Berisha[1,5,6], Mandana T. Manzari[1,9], Kristen Vogt [1,2,3], Oleg Gang [4,7,8] & Daniel A. Heller [1,2,3] ✉

Drug nanoaggregates are particles that can deleteriously cause false positive results during drug screening efforts, but alternatively, they may be used to improve pharmacokinetics when developed for drug delivery purposes. The structural features of molecules that drive nanoaggregate formation remain elusive, however, and the prediction of intracellular aggregation and rational design of nanoaggregate-based carriers are still challenging. We investigate nanoaggregate self-assembly mechanisms using small molecule fragments to identify the critical molecular forces that contribute to self-assembly. We find that aromatic groups and hydrogen bond acceptors/donors are essential for nanoaggregate formation, suggesting that both π-π stacking and hydrogen bonding are drivers of nanoaggregation. We apply structure-assembly-relationship analysis to the drug sorafenib and discover that nanoaggregate formation can be predicted entirely using drug fragment substructures. We also find that drug nanoaggregates are stabilized in an amorphous core-shell structure. These findings demonstrate that rational design can address intracellular aggregation and pharmacologic/delivery challenges in conventional and fragment-based drug development processes.

The spontaneous formation of therapeutic candidates and approved drugs into nanoscale aggregates can be deleterious to discovery processes but also can modulate pharmacologic properties[1]. Promiscuous aggregation of small molecules with proteins in solution contributes to false positive readouts in high-throughput screening and leads to off-target effects in cells[2,3]. Aggregates also non-specifically bind to proteins intracellularly and can lead to protein unfolding[4]. In addition, the mechanisms of sequestration of drugs into biomolecular condensates within cells is unknown[5]. Analytical strategies have been used to identify small molecule aggregation at the early stages of drug discovery[6–8]. However, the molecular structures that drive the formation of these promiscuous aggregations are difficult to predict[6].

On the other hand, drug carrier nanoaggregates (nanoparticles composed of a stabilized drug core) exhibit unique biological and pharmacologic behavior that can be exploited for therapeutic advantage. For example, drug carrier designs using nanoaggregates can alter the route of tissue or cellular uptake, serum retention, organ accumulation, and many aspects of pharmacokinetics and pharmacodynamics (PK/PD)[9–11]. To this end, drug nanoaggregates have been used for targeted delivery to overcome toxicity, increase bioavailability, and improve efficacy[12,13].

The chemical space available to address the increasing demand for new therapeutics against diverse and complex drug targets is exceedingly limited but may be improved by drug carrier

[1]Molecular Pharmacology Program, Memorial Sloan Kettering Cancer Center, New York, NY 10065, USA. [2]Graduate School of Medical Sciences, Weill Cornell Medicine, New York, NY 10065, USA. [3]Tri-Institutional PhD Program in Chemical Biology, Memorial Sloan Kettering Cancer Center, New York, NY 10065, USA. [4]Center for Functional Nanomaterials, Brookhaven National Laboratory, Upton, NY 11973, USA. [5]The Graduate Center of the City University of New York, New York, NY 10016, USA. [6]Department of Chemistry, Hunter College, City University of New York, New York, NY 10065, USA. [7]Department of Chemical Engineering, Columbia University, New York, NY 10027, USA. [8]Department of Applied Physics and Applied Mathematics, Columbia University, New York, NY 10027, USA. [9]Present address: Kaleidoscope Technologies, Inc., New York, NY 10003, USA. ✉e-mail: hellerd@mskcc.org

nanoaggregate formation[9]. Successful leads require not only potency and selectivity, but also safety and favorable PK/PD[14,15]. Although the concept of drug-likeness, which correlates chemical structures to absorption, distribution, metabolism, and excretion (ADME) properties, has helped drug discovery campaigns to identify safe and bioactive small molecule scaffolds at the preclinical stage, it also constrains the drug space[16]. Developing a parallel strategy for lead optimization so that potent, non-drug-like leads can be rescued as nanoaggregates may expand the usable repertoire of lead compounds in the clinic by modulation of their pharmacologic properties.

We previously described a colloidally stable nanoaggregate platform using an indocyanine dye (IR783) to encapsulate a wide range of small molecule drugs with unusually high drug loadings. Indocyanine dyes are amphipathic excipients that stabilize drug nanoaggregates[17]. We identified a method to predict drug nanoaggregates using molecular descriptors[17–20]. We established a quantitative structure-nanoparticle assembly prediction (QSNAP) model to predict the nanoparticle formation of a given drug[17]. We found that molecular features of a drug, particularly the number of high intrinsic state substructures (NHISS), a descriptor encompassing electron-withdrawing functional groups, predicted the formation of nanoaggregates with an indocyanine excipient[17]. However, like other self-assembled nanoaggregates, the internal structure and drivers of self-assembly are not known[6,8,21].

Here, we present a method to investigate drug aggregate self-assembly using fragment-based drug nanoaggregation assessments. We used small molecule fragments to directly compare chemical and structural features that promote nanoaggregate formation. We found that the self-assembly of hydrophobic compounds into nanoaggregates requires both π-π stacking and hydrogen bonding formation. We further confirmed these molecular parameters in a larger system using sorafenib, an approved kinase inhibitor drug, via chemical deletion of key functional groups. We also found that, unlike conventional drug nanocrystals, indocyanine dye-stabilized nanoaggregates of several approved drugs exhibit core-shell structures but lack intrinsic ordered internal packing, exhibiting an amorphous structure rather than a nanocrystal. This work provides chemical and structural insights to rationally guide the formulation of nanoaggregate drug carriers and to aid in circumventing promiscuous intracellular aggregate formation in lead identification.

## Results

### Biphenyl fragment small molecules serve as key scaffolds for nanoaggregate assembly

We developed a method to assess the self-assembly of molecular fragments into nanoaggregates. We added small molecule fragments dissolved in dimethyl sulfoxide (DMSO) dropwise into an aqueous solution containing an indocyanine dye, via adaptation of our previous method using standard drug molecules[17]. After centrifugation, we collected the pellets and resuspended them in water to assess the nanoaggregate morphologies (Fig. 1a). We established a method to determine which fragments form colloidally stable nanoaggregates versus precipitated drug, across a wide range of chemical properties. We set cutoffs wherein hydrodynamic diameters smaller than 500 nm and polydispersity index (PDI) less than 0.3 were considered nanoaggregates, as measured by dynamic light scattering (DLS). Samples with a diameter larger than 500 nm or PDI above 0.3 were considered precipitates and discarded. Samples with insufficient count rates in DLS were regarded as soluble, or below the critical aggregation concentration (CAC), a minimal concentration for self-assembly in solution[1,22].

We first assessed suitable scaffolds of fragments exhibiting the greatest potential to form nanoaggregates with a diverse substituent library. Since aromaticity is one of the most prevalent scaffolds in the current drug space and medicinal chemistry reactions[23], we examined hydrophobic fragment scaffolds with different aromatic ring counts, including benzene, naphthalene, and biphenyl scaffolds (Fig. 1b) with a carboxylic acid substituent (Fragment 1, 2 and 3). We formulated these fragments with two different indocyanine dyes - IR783 or indocyanine green (ICG); we included the latter because it is an FDA-approved molecule in the clinic. We then analyzed the degree of pelleting after centrifugation, indicating that a nanoaggregate may have formed (Fig. 1a). We observed three distinct results: a phenyl or mono-aromatic fragment formed a clear solution with no visible pellets, as expected from the high CAC of Fragment 1 (Supplementary Table 1 and 2). A naphthyl or fused-aromatic fragment (Fragment 2) formed visible pellets, but the system was not colloidally stable and settled quickly after resuspension. We speculate that the instability of fragment 2 is due to its rigid backbone and flat surface that result in its precipitation. The biphenyl fragment (Fragment 3) successfully formed a pellet and remained stable upon redispersing into a colloidal suspension (Fig. 1b).

We then characterized the size distribution of these nanoaggregates using DLS and atomic force microscopy (AFM). The naphthyl fragment formed a micron-size precipitate that rapidly settled after resuspension, whereas the biphenyl fragment formed much smaller particles with a 175 nm hydrodynamic diameter, falling within cutoff values (Fig. 1c). AFM showed that biphenyl fragment nanoaggregates exhibit a spherical morphology (Fig. 1d), similar to nanoaggregates composed of full-sized drugs[17].

Based on the size cutoff of these nanoaggregates, we tested additional common fragments to expand the array of scaffold candidates. We screened other polycyclic aromatic fragments like diphenylmethyl, diphenyl ether, and benzophenone scaffolds substituted with carboxylic acid (Fragments 4, 5, 6), which all resulted in nanoaggregate formation (Fig. 1e). Therefore, combing the results from solubility and size cutoffs, we concluded that fragment backbones with two non-fused aromatic rings provided the best scaffold to build our fragment library. To rationally compare the structural differences and efficiently extrapolate the key intermolecular forces involved in nanoaggregate formation among the groups of fragments, we proceeded with biphenyl fragments as our key scaffold due to its simple backbone with a diverse range of substituted analogues that enable direct comparisons. In addition, biphenyl is one of the most representative substructures in the current drug space, and it is assembled into smaller nanoaggregates among scaffold candidates[24].

### Fragment-based drug nanoaggregate assessment reveals the importance of hydrogen bonding in nanoaggregate self-assembly

We investigated the role of functional groups in nanoaggregate formation. As our previous study found that functional groups most important for nanoaggregation contained heteroatom double bonds[17], we attempted to form nanoaggregates with a small library of biphenyl fragments containing a variety of groups. We found that nanoaggregation propensity correlated with the presence of hydrogen bond donors and acceptors (Fig. 1f). We reasoned that heteroatom double bonds polarize the small molecule drugs and likely interact with neighboring molecules through hydrogen bonds.

We further analyzed the biphenyl fragments based on the presence of hydrogen bonding functional groups. First, we found that biphenyl scaffolds without hydrogen bond moieties formed micron size precipitates, likely due to high hydrophobicity (Fragment 17-19, Fig. 1f and Supplementary Table 1). Second, we found that nanoaggregate formation varied across different types of hydrogen bonds. Like many aniline compounds, 4-phenylaniline (Fragment 11) is a weak hydrogen bond acceptor; therefore we categorized it only as a hydrogen bond donor[25]. Fragments with only hydrogen bond donors or acceptors could not form nanoaggregates, but the presence of both donor and acceptor functional

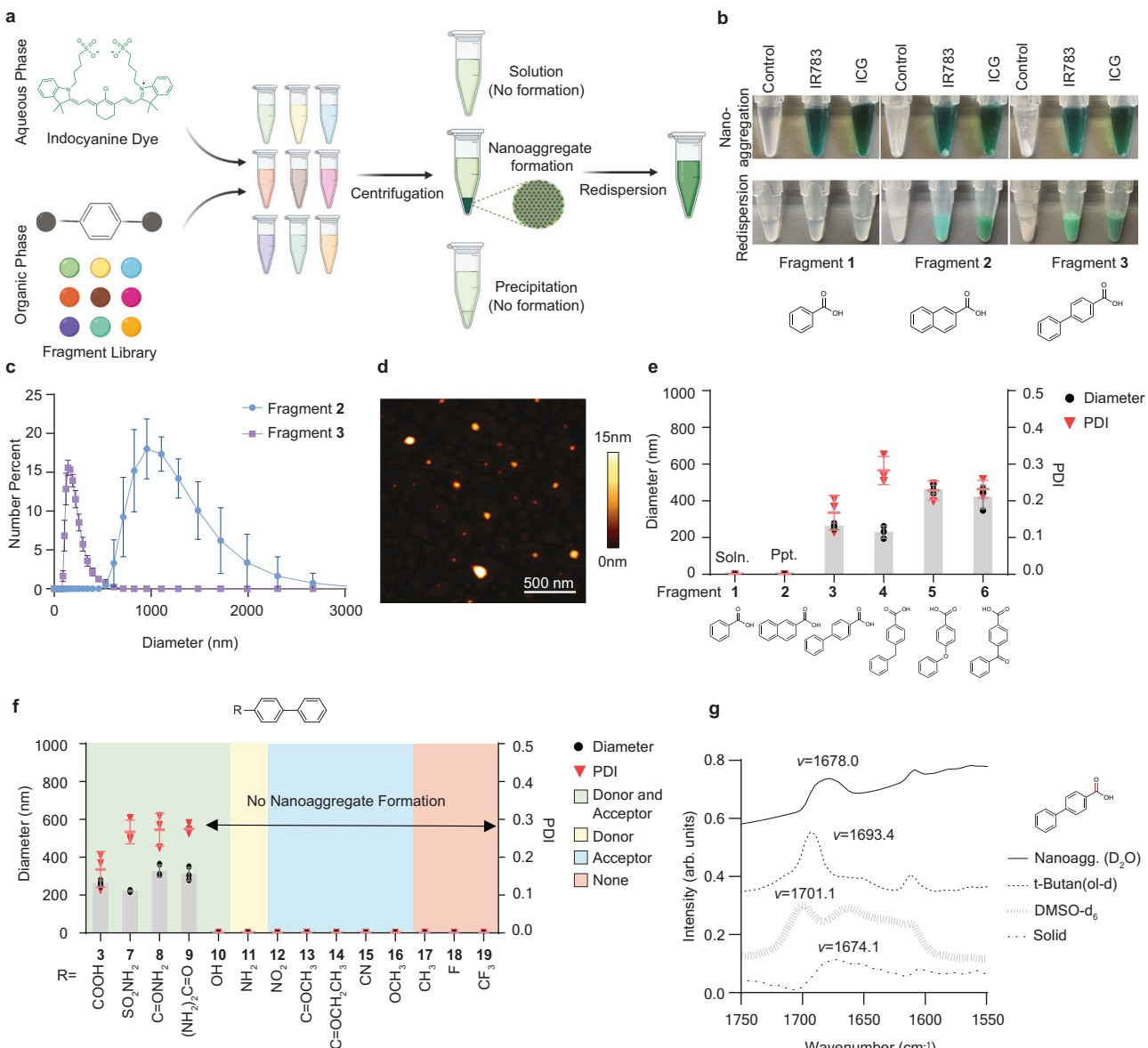

**Fig. 1 | Fragment-based drug nanoaggregation analysis reveals intermolecular hydrogen bonding moieties as important for nanoaggregate self-assembly.** **a** Schematic of fragment-based drug nanoaggregation assessment process with indocyanine excipients. **b** Results of redispersion of molecular fragments with Control (no excipient), IR783, or ICG excipients. The top row shows the pellet formation after centrifugation, and the bottom row shows the redispersion after removal of supernatant. **c** Size distribution of fragments **2** and **3** after pellet redispersion, via dynamic light scattering (DLS), N = 3 biological replicates. **d** Atomic force microscopy (AFM) image of biphenyl-4-carbocylic acid (Fragment **3**) nanoaggregates. Scale bar = 500 nm. At least 8 images were obtained with comparable results. **e** Average diameter of nanoaggregates formed using several aromatic backbone fragment scaffolds, measured by DLS, N = 3 biological replicates. PDI = polydispersity index. Soln. = fragment was soluble with indocyanine solution. Ppt. = precipitate. **f** Average diameter of nanoaggregates formed using biphenyl fragments with various functional group substituents measured by DLS, N = 3 biological replicates. **g** Fourier transform infrared spectra of the carbonyl group (in red) of fragment **3** in different solvents or upon redispersion of nanoaggregates. Peak wavenumbers ($v$) are listed. IR783 was used in (**c**–**g**). All bars are presented as mean values with error bars as the standard deviation. Source data are provided as a Source Data file.

groups resulted in nanoaggregate formation (Fig. 1f). We noticed one exception, however; while a hydroxyl group can also be a hydrogen bond donor and an acceptor, it failed to form nanoaggregates with IR783 (Fig. 1f). We surmise that this difference could be due to the capacity of heteroatom double bonds to undergo intermolecular double hydrogen bonding, whereas hydroxyl groups cannot. A similar trend in the size distribution, encapsulation efficiency, and loading was observed with nanoaggregates synthesized using ICG instead of the IR783 dye (Supplementary Fig. 1 and Supplementary Fig. 2). However, the hydroxyl substitution was more tolerated in ICG, and we observed formation of smaller nanoaggregates. This result is likely due to more extended

conjugation at the backbone of the ICG compared to IR783, such that a less stable intermolecular hydrogen bonding can be compromised through hydrophobic interactions.

We examined the type of hydrogen bonding in the nanoaggregates via chemical analysis. We used Fourier transform infrared spectroscopy (FTIR), focusing on the chemical environment of the carbonyl stretching region of the biphenyl-4-carboxylic acid (Fragment **3**, Fig. 1g). We observed a red-shift of the carbonyl stretching region of the fragment in the nanoaggregates, compared to a monomeric free acid of the fragment in a DMSO-d$_6$ solution[26]. The region shift also differed from that of the fragment dissolved in tert-butan(ol-d), a polar protic solvent that disrupts biphenyl-4-carboxylic

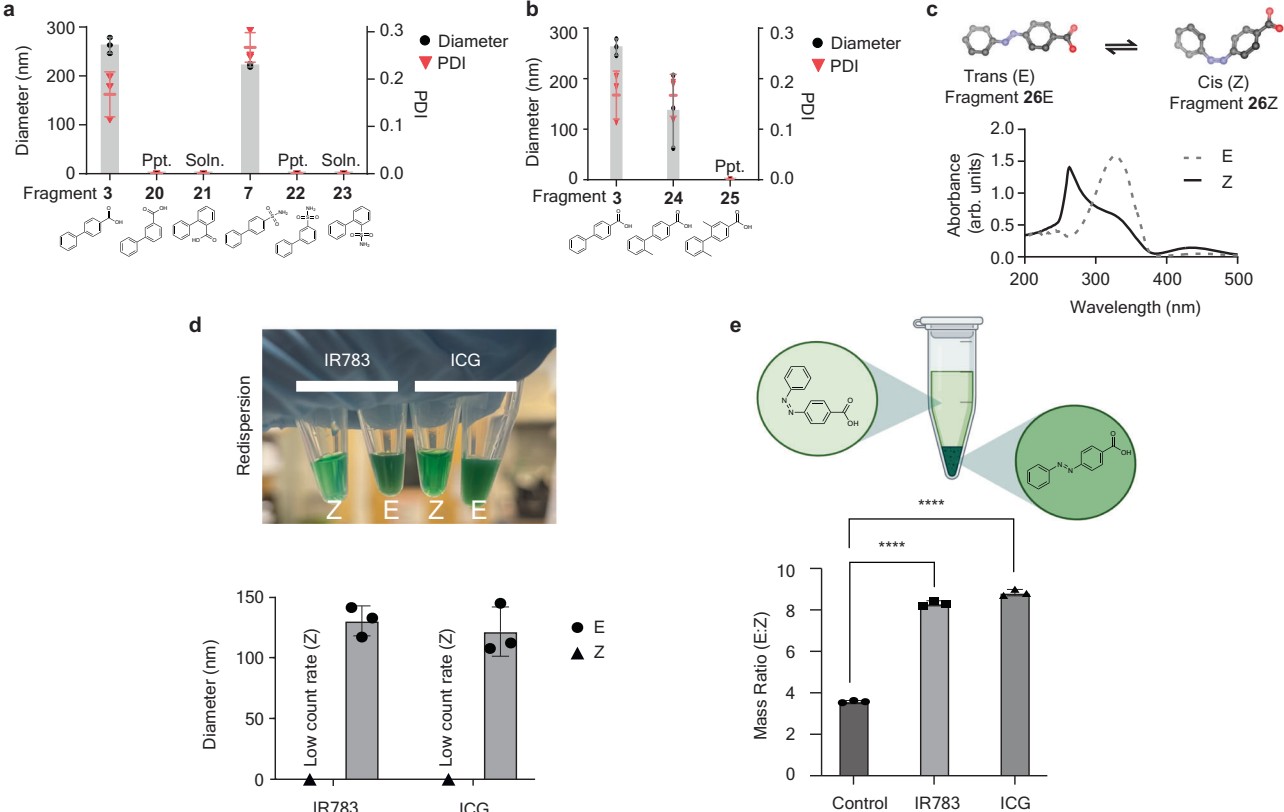

**Fig. 2 | Nanoaggregate self-assembly is facilitated by π-π stacking. a** Average diameter of regioisomerically substituted biphenyl fragment containing nanoaggregates, as measured by DLS. N = 3 biological replicates. **b** Average diameter of methyl-substituted biphenyl carboxylic acids containing nanoaggregates by DLS, N = 3 biological replicates. IR783 was used in (**a**, **b**). **c** Absorbance spectra of 4-phenyldiazenylbenzoic acid isomers. **d** Images of nanoaggregate suspensions of trans- and cis-4-phenyldiazenylbenzoic acid (top) and average diameter by DLS (bottom). N = 3 biological replicates. **e**, The mass ratio of trans to cis isomers in nanoaggregation without excipients (Control) or with IR783 and ICG, with starting materials of cis (Z) and trans (E) isomers in equal mass. ****P < 0.0001 (one-way ANOVA followed by Turkey's multiple comparisons test). N = 3 biological replicates. Soln. = fragment was soluble. Ppt. = fragment precipitated. All bars are presented as mean values with error bars as the standard deviation. Source data are provided as a Source Data file.

acid intermolecular dimers yet still partially forms hydrogen bonds with the solvent. Interestingly, the carbonyl stretching region of nanoaggregates resembles that of the solid powder, suggesting a similar chemical environment to a solid material, where the carboxylic acids arrange into homodimers[27]. These results suggest that intermolecular hydrogen-bond formation is one of the driving forces for nanoaggregate assembly.

**π-π stacking is a driving force for nanoaggregate self-assembly**

We aimed to further understand the role of aromaticity of drugs involved in the self-assembly process. As many small molecule drugs contain aromatic groups, we hypothesized that aromaticity plays an important role in the form of π-π stacking. To investigate, we formulated two groups of biphenyl fragments incorporating regioisomeric substitutions. We observed that only para-substituted biphenyls formed nanoaggregates (Fig. 2a and Supplementary Fig. 3a). Regioisomers had substantial differences in CAC despite identical calculated hydrophobicity values (CLogP or intrinsic solubility, Supplementary Table 2). The discrepancies are possibly due to the location of the functional groups, which can effectively change the steric effects, and are further affected by the 7% DMSO in the aggregation formation condition[28]. These conformational changes, resulting in different solvent exposure surface areas, cannot be predicted using calculated hydrophobicity values that rely on LogP computations of separated atoms or predefined fragments[29,30]. Ortho-substituted biphenyls were soluble in the aqueous solution used for synthesis (7% DMSO); as such, no

nanoaggregate was formed, and meta-substituted biphenyls precipitated instead of forming nanoaggregates. We believe that the preference of para-substituted biphenyls for nanoaggregate formation was due to steric hindrance and torsion angle preference in meta-substituted biphenyls but not para-substituted biphenyls, largely preventing π-π stacking[31].

We next investigated whether obviating π-π stacking could prevent the formation of nanoaggregates. We introduced methyl groups to biphenyl-4-carboxylic acid, which normally forms nanoaggregates. The addition of methyl groups drove precipitate formation rather than nanoaggregation (Fig. 2b and Supplementary Fig. 3b). We surmise that the addition of methyl groups increased the biphenyl torsional angles and eventually led to unfavorable configurations for nanoaggregation due to the prevention of a planar orientation of aromatic groups. Therefore, the para-substituted functional groups likely favored nanoaggregate formation due to the π-π stacking of the aromatic backbones.

To further examine the relationship between scaffold morphology and nanoaggregate formation, we used carboxylic acid-substituted azobenzene to modify the backbone orientation. Azobenzene is a well-studied photosensitive compound that can undergo trans-to-cis isomerization under UV light wherein the trans (E) isomer is more stable and has both aromatic rings parallel with each other, whereas the cis (Z) isomer is more labile with rings staggered on top of each other[32]. We drove the molecule from E to the Z isomers under excitation with 365 nm (UV) light for 12 hours and confirmed the formation by the increase of 265 nm and 440 nm

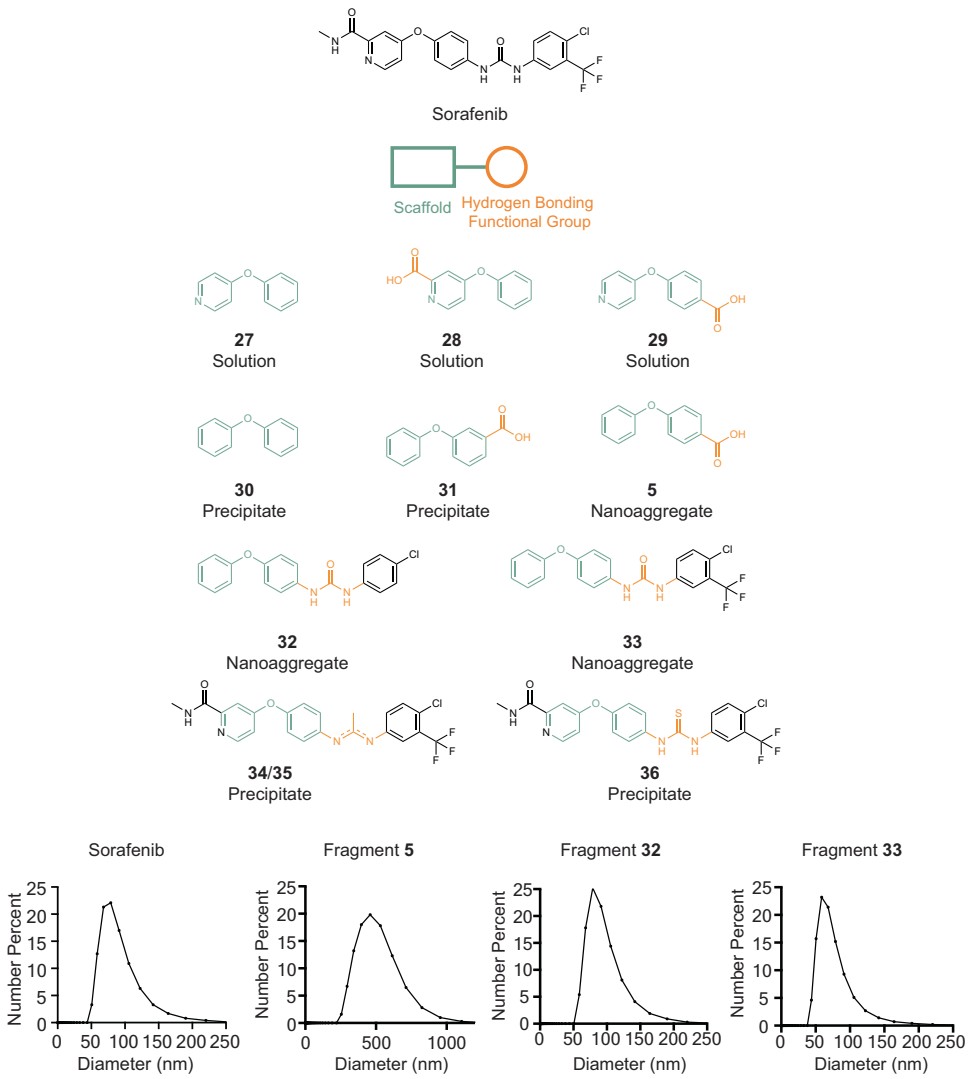

**Fig. 3 | Structure-assembly-relationship of sorafenib nanoaggregate formation.** Structures in green are core scaffolds for π-π stacking, and functional groups in orange are hydrogen bonding moieties. DLS data are shown for molecules that formed nanoaggregates with IR783 and are absent for those that do not. Molecules that precipitated or dissolved into the solution are denoted as Precipitate or Solution. Source data are provided as a Source Data file.

absorbance bands - two signature peaks for Z isomers (Fig. 2c). 39% of E isomers could still be detected, likely due to the reverse reaction driven by visible light and heat[32]. Next, we formulated both E and Z isomers of azobenzene with IR783 and ICG excipient dyes. Interestingly, only E isomer azobenzene formed nanoaggregates; Z-rich isomers produced low count rates via DLS, suggesting an undetectable concentration of nanoaggregates (Fig. 2d). Upon 6 hours of UV exposure of IR783-formulated E-isomer nanoaggregates, the PDI increased to above 0.3 (Supplementary Fig. 4a, b). After 12 hours of exposure, visible precipitates formed in the suspension, and a color change was visible, likely due to the isomerization and chemical changes of IR783 photobleaching (Supplementary Fig. 4c). We also examined if nanoaggregate assembly can self-sort one isomer over the other. By premixing both E and Z isomers before nanoprecipitation, the system preferentially sorted E isomers into nanoaggregates, as the concentration of the pellet and supernatant was measured upon column chromatography (Fig. 2e). The solubility of Z isomers further increased with the presence of IR783 and resulted in the isomeric preference in the nanoaggregation[33]. These results indicate that backbone orientation was critical for the formation of nanoaggregates.

## Structure-assembly-relationship approach reveals key functional components of nanoaggregate formation

We investigated the degree to which the above findings were conserved upon nanoaggregate formation of larger molecules such as approved drugs. We chose sorafenib, an FDA-approved kinase inhibitor, to determine how certain functional groups contribute to the intrinsic nanoaggregate formation with the IR783 dye[17]. Sorafenib-IR783 nanoaggregates exhibit a hydrodynamic diameter of $70.65 \pm 4.10$ nm and were colloidally stable in water for 7 days (Supplementary Fig. 5a, b), as well as in the pH range of 5.50–10.00 and salt concentrations under 5% NaCl (Supplementary Fig. 6).

We initially investigated the structural dependence of scaffolds for π-π stacking, and the position of hydrogen bonding moieties, on sorafenib-IR783 nanoaggregate formation. Based on our initial fragment-based studies above, we surmised that 4-phenoxypyridine (Fragment 27) in sorafenib, which resembles the biphenyl groups in a conjugated scaffold that promotes π-π stacking (Fig. 3). In sorafenib, hydrogen bond-forming functional groups are located at both positions 3 and 4 of the 4-phenoxypyridine moiety. However, the carboxylic acid-substituted 4-phenoxypyridine at position 3 (Fragment 28) or at position 4 (Fragment 29) did not form nanoaggregates

because they were soluble. Therefore, to increase the hydrophobicity of the core scaffold, we investigated diphenyl ether (Fragment **30**) as the π-π stacking scaffold and examined the significance of the hydrogen bond functional group positions. Like the meta-substituted biphenyl carboxylic acid (Fig. 2a, Fragment **20**), the carboxylic acid substituted 4-phenoxypyridine at position 3 (Fig. 3, Fragment **31**) did not form nanoaggregates. This result suggests that the N-methyl amide in sorafenib is not critical for hydrogen bond formation. Instead, the carboxylic acid-substituted diphenyl ether at position 4 (Fragment **5**) formed nanoaggregates (Fig. 3), associating the location of hydrogen bonding to the assembly of sorafenib nanoaggregates.

To further evaluate whether N-methyl amide affects nanoaggregate formation in sorafenib, we synthesized an analogue (Compound **32**, Supplementary Fig. 7a, b) retaining the key backbone moieties of sorafenib. Nanoaggregates formed with this analogue exhibited a strikingly similar size profile compared to those of sorafenib, suggesting that the essential molecular interactions contributing to sorafenib nanoaggregate formation are recapitulated in this fragment and that the urea moiety likely plays a key role in nanoaggregate formation through intermolecular hydrogen bonding.

We investigated the importance of fluorine in sorafenib nanoaggregate formation. The molecular descriptor NHISS identified fluorine as an important functional group for nanoaggregate formation[17]. We further synthesized another sorafenib analogue (Compound **33**, Supplementary Fig. 7c, d) by adding the trifluoromethyl group. The compound successfully formed nanoaggregates, albeit with a smaller size compared to Compound **32** (Fig. 3). These results suggest that fluorine does not contribute to the major driving force for nanoaggregate formation; rather, it may stabilize the nanoaggregates by reducing the overall size.

To confirm the key driving force for nanoaggregate formation is the hydrogen bond from the urea moiety in sorafenib, we utilized Compound **34** and **35** (structural isomers) to reduce the urea to imine, and locally eliminate the hydrogen bonding ability. We observed that these compounds no longer formed nanoaggregates. Further replacing urea with thiourea, a weak hydrogen bond acceptor, in sorafenib (Compound **36**) generated a large PDI precipitation, suggesting a highly heterogenous aggregates with IR783 (Supplementary Table 1), and the importance of hydrogen bonding in sorafenib nanoaggregation.

Overall, the results showed that assembly of sorafenib nanoaggregates requires hydrogen bonds within the urea functional group, and the conjugation system of diphenyl ether or 4-phenoxypyridine to provide a backbone scaffold. Hence, through the reconstruction of a more complex small molecule drug, we have determined key parameters underlying formation of sorafenib nanoaggregates.

### Indocyanine and small molecule self-assembly produces amorphous core-shell nanoaggregates in solution

In order to visualize the hydrogen bonding and π-π interacting motifs in sorafenib-IR783 nanoaggregates, we ran an all-atom molecular dynamics (AAMD) simulation. The 200 ns simulation consisted of four IR783 molecules and twelve sorafenib molecules in a box with explicit water. The simulation reached an initial equilibrium at around 20 ns, based on calculation of the average molecular distance (Supplementary Fig. 8). In addition, we plotted a kymograph showing the time course of every hydrogen bonding interaction during the 200 ns simulation (Supplementary Fig. 9a). We identified 186 unique hydrogen bonding at the conclusion of the 200 ns simulation. By fitting the kymograph to an exponential plateau curve, the function reached an asymptote of 195.4 with 95% CI [189.8, 199.5], suggesting that the simulation reached approximately 95% of the maximal number of hydrogen bond interactions within 200 ns (Supplementary Fig. 9b). Therefore, the results suggest stabilization of the simulated nanoaggregate structure in that timeframe. All four IR783 molecules were

located at the surface of the nanoaggregates, and the sorafenib molecules largely localized away from the solvent (Fig. 4a). Although the internal arrangement of sorafenib molecules was largely disordered, we observed clear indications of hydrogen bonding from the urea moiety, and potential π-π interactions (Fig. 4a). We analyzed the interactions of key hydrogen bond-forming functional groups throughout the simulation by calculating the formation of each type of hydrogen bond between sorafenib molecules. We found that intermolecular hydrogen bonds from the urea moieties between two sorafenib molecules occurred at the highest probability as compared to other types (Fig. 4b). We also quantified likely π-π interactions at every 10 ns during the simulation timeframe, and we measured a centroid distance from 3 Å – 5 Å to include parallel stacked, parallel displaced, edge-to-face and T-shaped interactions[34,35]. We found that, on average, 77% (±17.9%) of molecules fit these molecular distance and orientation criteria that would permit π-π interactions (Fig. 4c and Supplementary Fig. 10). At each frame, occasional edge-to-face π-π interactions (two or three interactions per frame) were observed, but the majority of the π-π interactions were parallel stacked or parallel displaced. Overall, we observed that π-π interactions tended to be more transient as compared to intermolecular hydrogen bonding in the simulation, possibly owing to the relatively weak nature of π-π interactions[36,37]. These molecular simulation results support the conclusion that intermolecular hydrogen bonds of the urea functional groups and π-π stackings were dominant interactions in the sorafenib-IR783 nanoaggregates.

Next, we investigated the internal structure of the nanoaggregates, as these details are largely unknown for drug aggregates[17,20]. Therefore, we performed X-ray scattering and electron microscopy to investigate the morphology and internal structure of the nanoaggregates. Transmission electron microscopy (TEM) images and fast Fourier transformation (FFT) showed that sorafenib-IR783 nanoaggregates were spherical (Fig. 4d) with an amorphous structure (Fig. 4e). This was further confirmed by X-ray scattering analysis of the nanoaggregates as lyophilized samples, where multiple broad oscillations were observed at the high q range (>0.1 Å$^{-1}$) (Fig. 4e). Compared to the powders of sorafenib and IR783 alone, which exhibited crystallinity via wide angle scattering, the lyophilized sorafenib nanoaggregates exhibited a wide halo peak indicating an amorphous solid. The scattering features in the small angle region indicate aggregation that was likely caused by lyophilization (Fig. 4e), in which several broad oscillations at the intermediate q range (0.01–0.05 Å$^{-1}$) indicated non-uniform aggregations with large sizes. On conducting X-ray scattering analysis of sorafenib-IR783 nanoaggregates in water, we similarly observed amorphous features (Fig. 4e). The radius of gyration ($R_g$ = 250 ± 63.4 Å) of sorafenib-IR783 nanoaggregates is comparable to its hydrodynamic radius, measured by DLS (35.32 ± 2.05 nm, Supplementary Fig. 11).

Next, we examined the core-shell arrangement of drug and dye molecules in the nanoaggregates. First, we compared the nuclear magnetic resonance (NMR) of the monomeric sorafenib, IR783, and their nanoaggregates in solution (Supplementary Fig. 12). Compared to the ¹H NMR spectra of sorafenib, IR783 monomers, the spectrum of nanoaggregates revealed an overall reduction in the proton signals from aromatic, methine groups and secondary amines, confirming aggregation of the monomers[38]. We found that only two signal peaks (1.62 and 2.58 ppm) were quantifiable in sorafenib-IR783 nanoaggregates. These two peaks were likely from the alkyl chain between the sulfate and the tertiary amine in IR783, suggesting the alkyl chain is solvent exposed. Additionally, the molecular simulation of the nanoaggregates also showed a decreased solvent-accessible surface area overtime for sorafenib but remained unperturbed for IR783 (Supplementary Fig. 13). These results indicate that IR783 constitutes the outer shell of sorafenib nanoaggregates and sorafenib is entrapped within a solid core,

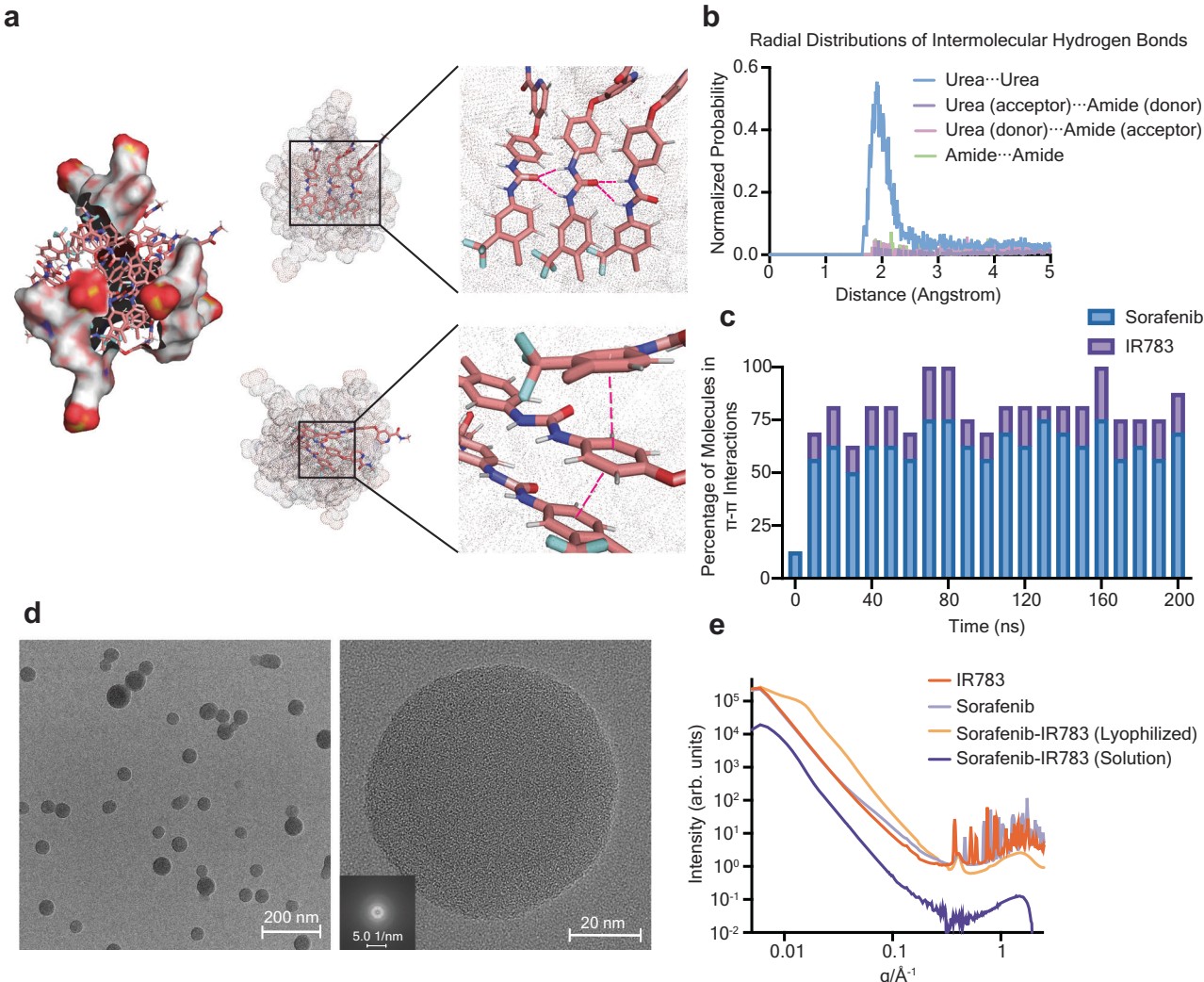

**Fig. 4 | Internal structure of sorafenib nanoaggregates. a** Molecular dynamics simulation of sorafenib-IR783 nanoaggregate. IR783 molecules are drawn in surface and sorafenib molecules are drawn in sticks. Representative hydrogen bonds in simulation are shown at the top, and representative π-π stackings are shown at the bottom. **b**, Probability of common types of intermolecular hydrogen bond formation during the 100 ns simulation. **c** Percentage of the total number of molecules involved in π-π interactions during the course of simulation. **d** Representative TEM images of sorafenib-IR783 nanoaggregates at lower (58 kx, left) and higher (630 kx, right) magnifications with the fast Fourier transformed image (inlet). 5 images were obtained with comparable results. **e** Small and wide-angle X-ray scattering of the sorafenib, IR783 powders, lyophilized nanoaggregates, and nanoaggregates in water. The analysis to obtain the radius of gyration ($R_g = 250 \pm 63.4$ Å) from form factor fitting for the nanoaggregates is shown in Supplementary Fig. 11. Source data are provided as a Source Data file.

similar to other excipient sorafenib nanoaggregates observed in other works[39].

We applied energy-dispersive X-ray spectroscopy (EDS) coupled with the high angle annular dark field-scanning transmission electron microscopy (HAADF-STEM) to analyze the composition and spatial distribution of drug and dye molecules in the nanoaggregates (Fig. 5). IR783 and sorafenib were identified by their characteristic sulfur and fluorine elements, respectively. The elemental mapping of sorafenib nanoaggregates suggested that sorafenib largely clustered towards the center of the nanoaggregate, while IR783 distributed more homogeneously. The same experiment was also performed on trametinib-ICG and regorafenib-ICG nanoaggregates that also fit the size and stability criteria set to denote nanoaggregate formation (hydrodynamic diameter of $46.84 \pm 12.18$ nm and $76.07 \pm 3.72$ nm respectively, and colloidal stability for at least 3 days (Supplementary Fig. 5). Trametinib, regorafenib, and ICG were identified by iodine, chlorine and sulfur, respectively (Fig. 5). Multiple regorafenib-ICG nanoaggregates appear in the HAADF-STEM image, thus resulting in the observed morphology (Fig. 5a). Similar to sorafenib, the trametinib

and regorafenib nanoaggregates also exhibited a segregated distribution of the drug and dye, suggesting a core-shell structure involving hydrophobic drugs encapsulated by amphiphilic dye molecules.

## Discussion

Herein, we investigated the relationship between small molecule chemistry and nanoaggregate assembly using a fragment-based approach to uncover the key intermolecular forces driving assembly of nanoaggregates. We discovered that hydrogen bonding, particularly with both donors and acceptors on the same functional group, and π-π stacking, are critical for nanoaggregate formation. Furthermore, we determined how these principles of nanoaggregate assembly, discovered using small fragments, can be extended to full-size drug compounds through structure-assembly relationship. Finally, we used molecular dynamics simulation, STEM-EDS, NMR, and X-ray scattering to analyze the nanoaggregate structures, which exhibit a largely amorphous structure consisting of a drug core and dye shell. These results confirmed that nanoaggregate assembly is driven by multiple molecular forces (i.e., π-π stacking and hydrogen bonding),

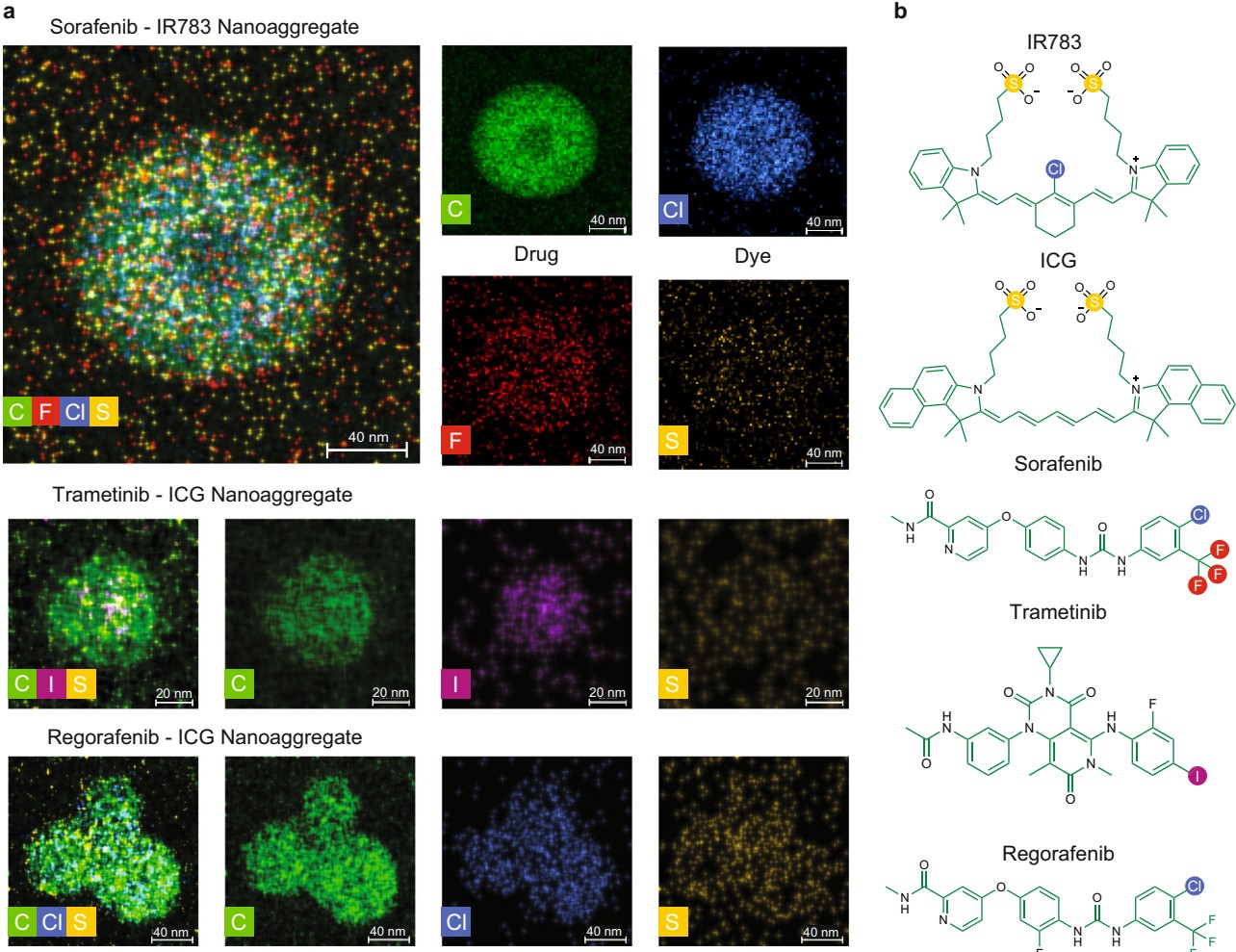

**Fig. 5 | Molecular distribution within nanoaggregates. a** Representative EDS elemental mapping of nanoaggregates composed of different kinase inhibitors with details of each element. At least 10 images per nanoaggregate type were obtained with comparable results. **b** The elements used in the analysis highlighted in the chemical structures.

which resulted in non-uniaxial interactions of the drug and dye molecules.

Systematic investigations of the intermolecular forces for encapsulated small molecules revealed insights for nanoaggregate formation. Hydrogen bonds serve as a driving force to stabilize the solid core of the nanoaggregates. Dimerization through intermolecular hydrogen bonding could potentially stabilize molecular packing in the nanoaggregates. Some previous studies focused on the interactions between excipients and drugs during the nanoaggregate formation, suggesting the interactions are mainly hydrophobic or electrostatic[40,41]. However, in systems like indocyanine-stabilized nanoaggregates, with a drug-to-dye stoichiometry ratio of near 10:1, drug-drug interactions can play a dominant role in nanoaggregate formation and are often overlooked. In addition to the interaction with the excipients, dimerization of the drug through hydrogen bonds explains a typical high encapsulation efficiency of drugs across the dye-stabilized nanoaggregates[17].

We note that the fragments produced nanoaggregates with varying hydrodynamic diameters. We surmise that the size relates to nanoaggregate stability conferred by the drug and dye structures. We note that our previous work found that nanoparticle size correlated with some accuracy ($R^2 = 0.84$, 95% CI [0.22, 0.98]) to a molecular descriptor that included an electronegativity term[17], suggesting some relationship with certain functional groups like hydrogen bonding

moieties, potentially analogous to the relationship of nanoaggregate stability conferred by these groups.

Similar molecular interactions to those described herein were found in surfactant micelles, where π-π stacking and hydrogen bonding also play important roles in the self-assembly of liquid colloids[42–46]. A polar group or a hydrophilic group usually remains hydrated in surfactant micelles[43,44], but we discovered that these polar and hydrophilic functional groups can also stabilize the solid colloids via intermolecular dimerization, and potentially phase-separated from the liquid. Intermolecular dimerization through hydrogen bonding has also been observed in certain supramolecular dendrimers, where monomeric dendrons can self-assemble into a hexametric rosette through carboxylic acid dimerization[47,48]. In addition, π-π stacking as a self-assembly mechanism has been shown in peptide-based drug delivery cargos, where aromatic amino acids like phenylalanine or tryptophan can stack to the aromatic moieties in drugs or nucleic acids[49,50]. Similarly, we found that indocyanines dyes, which include extensive conjugation and aromaticity, can also localize largely to the surface of the nanoaggregates, facilitating colloidal stability.

Additional findings were also discovered in the nanoaggregate structures. Previous works suggest that these colloidal nanoaggregates have filled and non-hollow structures, as opposed to polymeric micelles[22,51]. Our results added to this knowledge, wherein we found that dye-stabilized nanoaggregates exhibit core-shell structures

that are also structurally amorphous (no drug or dye crystallinity). However, whether the structure of nanoaggregates is specific to dye-stabilized structures requires further investigations on other excipients.

Drugs with poor solubility, particularly those classified as Class II or IV pharmaceuticals, require extensive formulation development to enhance their absorption, facilitate passage across biological barriers, and maintain efficacious drug concentrations in the body[52]. Nanoaggregates with amorphous structures, which can offer large surface areas, can thereby raise the dissolution rates of these drugs while preventing undesired precipitation in biological fluids[17,52,53]. Previous research has demonstrated that colloidal nanoaggregates exhibit superior stability in serum compared to their freed drug counterparts, and the use of amorphous dispersions effectively enhanced plasma drug exposure[54,55].

Extensive studies of the chemical features of small molecule drugs that drive nanoaggregate formation are crucial for drug discovery and delivery approaches. The presence or absence of these features in a lead compound can provide predictions for aggregation that hinders drug development processes, or nanoaggregate formation that can facilitate delivery. Regarding the latter, drug features, that enable π-π stacking and intermolecular hydrogen bonding, can translate to nanomedicine development, potentiating the modulation of ADME properties separate from the function of a molecule, which can potentially expand the drug space by permitting expanded structural diversity while separately preserving sufficient pharmacologic parameters to enable administration into humans[9].

## Methods

### Materials and reagents
Small molecule fragments or starting materials were purchased from Sigma Aldrich (St. Louis, MO), Thermo Fisher Scientific (Tewksbury, MA), AA Blocks Inc. (San Diego, CA), AK Scientific Inc. (Union City, CA), Enamine (Monmouth Jct., NJ), ChemBridge (San Diego, CA) (Supplementary Table 1). Fragment **31-33** were synthesized at Wuxi AppTec (Shanghai, China). sorafenib, trametinib and regorafenib were purchased from MedChemExpress LLC. Indocyanine green (ICG or IR125) was purchased from Fisher Scientific. All other reagents were purchased from Sigma Aldrich.

### Nanoaggregate synthesis and characterizations
We prepared 20 mg mL$^{-1}$ small molecules in DMSO, and dropwise added 50 μL over vortex into a 650 μL 0.73 mg mL$^{-1}$ IR783 (or ICG) aqueous solution. The solution was then centrifuged (30,000 g, 15 min), and the supernatants were separated from the pellets (if any). The pellet was re-suspended in 200 μL of double distilled water. Dynamic light scattering (Zetasizer Nano ZS, Malvern) was used to obtain the size and polydispersity of the nanoaggregates.

### Atomic force microscopy
Biphenyl-4-carboxylic acid nanoaggregates were diluted 1:10 with 20 mM MgCl$_2$. 40 μL of the nanoaggregates in MgCl$_2$ were deposited onto freshly cleaved mica (Pelco Mica Disc, V1, Ted Pella) for 15 min. After the incubation, the sample was rinsed with 1 mL deionized water and the surface was dried using a stream of argon. AFM images were captured using an Nanowizard V (JPK Bruker) microscope in AC Mode Imaging at room temperature. AFM probe with resonance frequencies of approximately 75 kHz and a spring constant of 3 N m$^{-1}$ was used for imaging. Images were collected at a speed of 3 Hz with an image size of 2 × 2 μm at 512 × 512 pixels resolution. The images were processed with JPK Data Processing software.

### Fourier transform infrared spectroscopy
Redispersion of nanoaggregates were prepared in D$_2$O and 4-phenylbenzoic acid were prepared in t-butan(ol-6) and DMSO-d6 to test solvent effects on spectral shifts. Solutions were prepared at 20 mg mL$^{-1}$, and 5 μL of solution was placed between two calcium fluoride windows, with a six-micron spacer between windows. The solid state 4-phenylbenzoic acid sample was measured by dissolving the drug in THF at 20 mg mL$^{-1}$ and allowing 5 μL to dry on a single calcium fluoride window. The FTIR spectra were acquired in a Bruker Vertex 70 spectrometer (Bruker Optik GmbH, Germany) with a spectral resolution of 8 cm$^{-1}$, and a range of 4000 cm$^{-1}$ to 800 cm$^{-1}$. Bruker OPUS 7.2 software was used for sample acquisition.

### Isomerization of 4-phenyldiazenylbenzoic acid
4-phenyldiazenylbenzoic acid (Sigma Aldrich) was dissolved in DMSO (20 mg mL$^{-1}$) and placed on a stir plate. A 365 nm handheld UV lamp (Crystal Technologies, The BioGlow series, 12 W) was used for isomerization, and the product was monitored in a UV-VIS-NIR spectrophotometer (Jasco 670) and quantified in C18 (150 mm × 2.1 mm internal diameter, 3.5 μm; Agilent Technologies) analytical column using a mobile phase of acetonitrile and deionized water, both in 0.1% trifluoroacetic acid. The gradient from 5% to 90% acetonitrile in 5 min, 90% to 95% acetonitrile in 3 min, and flow at 1 mL min$^{-1}$ showed a retention time of 4.4 min for Z isoform and 5.4 min for E isoform.

### Critical aggregation concentration and calculate hydrophobicity
Fragments for small molecule drugs were prepared in 20 mg mL$^{-1}$ (DMSO) and dropwise added 50 μL over vortex into a 650 μL of water. Samples were bath-sonicated for 3 mins before centrifugation (30,000 g, 30 min), and the supernatants were separated from the pellets (if any). The supernatants were collected and quantified in C18 analytical column. Calculated hydrophobicity (CLogP and intrinsic solubility) was calculated using ChemAxon's Chemicalize platform.

### Turbidity assessments of sorafenib-IR783
Normalized turbidity measurement sorafenib-IR783 nanoaggregates were aliquoted and redispersed in a range of pH buffer conditions, and a range of salt concentrations using NaCl. Turbidity was measured using absorbance at 600 nm and normalized to each buffer condition without nanoaggregates. Normalized turbidity was calculated using turbidity in pH = 7.4 or water with 0% NaCl as a standard, respectively. N = 3 biological replicates were performed.

### Synthesis of sorafenib analogues
Reactions were performed in oven-dried glassware under air and at room temperature with magnetic stirring. TLC was performed on 0.25 mm E. Merck silica gel 60 F254 plates and visualized under UV light (254 nm). Silica flash chromatography was performed on E. Merck 230–400 mesh silica gel 60. NMR spectra were recorded on Bruker UltraShield Plus 500 MHz instruments at 24 °C in DMSO-d6 unless otherwise indicated. Chemical shifts are expressed in ppm relative to TMS (1 H, 0 ppm); coupling constants are expressed in Hz.

4-Chlorophenyl isocyanate or 4-Chloro-3-(trifluoromethyl)phenylisocyanate (0.1 mmol, 1eq) and 4-Phenoxyaniline (0.1 mmol, 1eq) were dissolved in anhydrous DCM (3 ml) for 15 min and followed by adding TEA (0.02 mmol, 0.2eq) drop by drop. The result reaction was stirred overnight at room temperature. The reaction was monitored by TLC plates. The result crude product was concentrated and purified by silica flash chromatography to give 1-(4-Chlorophenyl)-3-(4-phenoxyphenyl)urea (Fragment **29**) or 1-[4-Chloro-3-(trifluoromethyl)phenyl]-3-(4-phenoxyphenyl)urea (Fragment **30**) as white powders.

**Fragment 29:** White solid (73%, 1 step). **¹H NMR** (500 MHz, DMSO-$d_6$) δ 8.78 (s, 1H), 8.69 (s, 1H), 7.51 – 7.43 (m, 4H), 7.39 – 7.28 (m, 4H), 7.08 (tt, J = 7.4, 1.1 Hz, 1H), 7.02 – 6.89 (m, 4H). **¹³C NMR** (126 MHz, DMSO-$d_6$) δ 157.58, 152.48, 150.77, 138.74, 135.47, 129.89, 128.58, 125.25, 122.76, 120.06, 119.72, 119.67, 117.61 (Supplementary Fig. 7).

**Fragment 30:** White solid (61%, 1 step). $^1$H NMR (500 MHz, DMSO-$d_6$) δ 9.14 (s, 1H), 8.85 (s, 1H), 8.11 (d, J = 2.4 Hz, 1H), 7.67 – 7.57 (m, 2H), 7.55 – 7.45 (m, 2H), 7.43 – 7.32 (m, 2H), 7.09 (t, J = 7.4 Hz, 1H), 7.04 – 6.89 (m, 4H). $^{13}$C NMR (126 MHz, DMSO-$d_6$) δ 157.53, 152.49, 151.13, 139.42, 135.10, 131.95, 129.91, 123.00, 122.84, 120.48, 119.66, 117.71 (Supplementary Fig. 7).

## Molecular dynamics simulations and analysis

Topologies of IR783 and sorafenib were generated using the general Amber forcefield. The forcefield was selected as it is recommended for small hydrophobic molecules. Amber topologies were converted to Gromacs format using Parmed to run GPU-accelerated simulations on Gromacs. All-atom molecular dynamics (AAMD) simulations were run with explicit solvent, using the TIP3 model, at neutral conditions. Twelve molecules of sorafenib and four molecules of dye were placed randomly in a five-nanometer box. This ratio was chosen to match the experimental molar equivalents of drug and dye prior to mixing. The energy of the system was minimized to ensure that there were no steric clashes. NVT (constant number of atoms, volume, and temperature) and NPT (constant number of atoms, pressure, and temperature) equilibration was conducted for 100 ps to equilibrate solvent molecules around the dye and drug. The equilibrated system was run for 200 ns and the coordinates were saved every 200 ps, for a total of 1000 frames.

We used the CPPTRAJ toolkit on AmberTools to analyze the most types of common hydrogen bonds and construct radial distribution plots of specific hydrogen bonds. Intermolecular interactions were excluded from the analysis. The values were normalized by the density of 0.033456 molecules/angstrom$^3$, which corresponds to a density of water approximately equal to 1.0 g mL$^{-1}$. Since the radial distribution functions do not incorporate an angle cut-off, the distance distributions include probabilities of hydrogen bonds at any angle. All frames of the trajectory were used in the analysis.

Simulation trajectories at every 10 ns frame were saved as pdb files. For π-π interactions between sorafenib molecules, only sorafenib molecules were extracted from the trajectories, and a build-in function for π-π interaction analysis using Discovery Studio Visualizer (BIOVIA) was applied to select for all possible interactions (face-to-face, offset, and T-shaped interactions). The number of the interactions were counted manually for each frame. The procedure was also applied to count sorafenib-IR783 π-π interactions.

The GROMACS in-built trajectory analysis function (gmx sasa) was used to calculate solvent accessible surface area.

## Energy dispersive x-ray spectroscopy analysis

HAADF-STEM imaging and STEM-EDS elemental mapping of the drug nanoaggregates were acquired with a Thermo Fisher Scientific Talos F200X at an accelerating voltage of 200 kV. For sample preparation, 200-mesh ultrathin carbon film Au grids (Electron Microscopy Sciences) were glow discharged at 5 mA for 10 sec. Next, drug nanoaggregates (0.02-0.05 mg mL$^{-1}$) were deposited onto the glow-discharged grids for 1–1.5 minutes, and the residual liquid was removed using a piece of filter paper. The grids were washed by deionized water and dried overnight at room temperature prior to imaging.

## Small- and wide-angle x-ray scattering analysis

SAXS/WAXS scattering data of sorafenib, IR783, and sorafenib nanoaggregates in solution/lyophilized were collected at the Life Sciences X-ray Scattering beamline (LiX, 16-ID) at the National Synchrotron Light Source II (NSLS-II) at Brookhaven National Laboratory (BNL). LiX utilizes an undulator source and a Si(111) monochromator and data was collected on 3 Pilatus detectors (SAXS: Pilatus 1 M, 2 offset WAXS detectors: Pilatus 300 K)[56]. Microbeam scattering with a beam size of ~5 μm was used to measure the solid samples where Kapton was used

as a reference and subtracted from the samples. For each sample exposed to the X-ray beam, 10 frames with an exposure time of 1 second were collected. The data was merged, averaged, subtracted, and packed into HDF5 format using our in-house py4xs software[57]. Solution scattering with a beam of ~400 μm was used to measure the sorafenib nanoaggregate (1 mg mL$^{-1}$) in water. The samples were loaded in an in-house solution scattering box housing a movable 3-channel flow cell, and for each sample, 5 frames with an exposure time of 1 second were collected. Data was processed using the py4xs software and water was used as reference and subtracted from the samples. Igo Pro 8 (WaveMetrics) and built-in Irena package was used to analyze the solution scattering data, where a spherical size distribution model was applied to obtain the radius of gyration ($R_g$).

## Reporting summary

Further information on research design is available in the Nature Portfolio Reporting Summary linked to this article.

## Data availability

All data generated in this study are provided in the Supplementary Information and Source Data file. Source data are provided with this paper.

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

## Acknowledgements

This work was supported in part by the, NCI (R01-CA215719 and Cancer Center Support Grant P30-CA008748), NINDS (R01-NS116353, R01-NS122987), NIDDK (R01-DK129299), the National Science Foundation CAREER Award (1752506), the Department of Defense Congressionally Directed Medical Research Program

(W81XWH-21-1-0188), the American Cancer Society Research Scholar Grant (GC230452), the Pershing Square Sohn Cancer Research Alliance, the Expect Miracles Foundation - Financial Services Against Cancer, the Louis and Rachel Rudin Foundation, Emerson Collective, the Alan and Sandra Gerry Metastasis Research Initiative, Mr. William H. Goodwin and Mrs. Alice Goodwin and the Commonwealth Foundation for Cancer Research, and the Experimental Therapeutics Center of Memorial Sloan Kettering Cancer Center. N.B. was supported by the Tow Foundation Predoctoral Fellowship, Center for Molecular Imaging and Nanotechnology at MSKCC. We also thank the support from the Center for Functional Nanomaterials at BNL, the BNL Laboratory Directed Research and Development grant, and the Office of Science, Office of Basic Energy Sciences, of the U.S. Department of Energy under Contracts No. DE-SC0012704 and No. DE-AC02-05CH11231. LiX beamline is part of the Life Science Biomedical Technology Research resource, co-funded by the National Institute of General Medical Sciences (NIGMS) under grant P41 GM111244 and by the DOE Office of Biological and Environmental Research under grant KP1605010, with additional support from NIH under grant S10 OD012331. The operation of NSLS-II is supported by the U.S. Department of Energy, Office of Basic Energy Sciences, under contract No. DE-SC0012704. O.G. is partially supported by the US Department of Energy, Office of Basic Energy Sciences, Grant DE-SC0008772. K.V. acknowledges the support provided by the National Science Foundation Graduate Research Fellowships Program (1746886) and T32 Training grants (GM115327 and GM136640). Finally, we thank Dr. Sooyeon Hwang at the Center for Functional Materials for the support on Talos operation and EDS image collection; we thank Drs. James Byrnes, Shirish Chodankar, and Lin Yang for the discussion and support on the LiX results and data collection; we thank the molecular cytology core facility at MSKCC for AFM measurements and the NMR analytical core facility for NMR measurements. Figure 1a and Fig. 2e were created with BioRender.com.

## Author contributions

C.C., M.M., D.A.H., conceived the research. C.C., Y.W., S.W., N.B., O.G., and D.A.H. designed experiments and analyzed the data. C.C., Y.W., S.W., N.B., M.M., and K.V. performed experiments. C.C. and D.A.H. wrote the manuscript. S.W. edited the manuscript. D.A.H. and O.G. supervised the research.

## Competing interests

D.A.H. is a cofounder and officer with equity interest of Lime Therapeutics Inc., cofounder with equity interest of Selectin Therapeutics, Inc. and Resident Diagnostics, Inc., and a member of the scientific advisory board of Concarlo Therapeutics, Inc., Nanorobotics Inc., and Mediphage Bioceuticals Inc. The remaining authors declare no competing interests.
