## [Peer Review File · Nature Communications]

REVIEWER COMMENTS

Reviewer #1 (Remarks to the Author):

The manuscript by Chen et al. investigates the structural features of molecules that allow formation of nanoaggregates using small molecule fragments and structure-assembly-relationship analysis. The authors first evaluate aromatic small molecule fragments for self-assembly with indocyanine excipients and demonstrate the importance biphenyl structures and the presence of hydrogen bonding and π - π stacking interactions for nanoaggregation. Investigation of structural analogs of a known nanoaggregator, sorafenib, further supported the importance of these structural features through structure-assembly-relationship analysis. Molecular dynamics simulations support the existence of hydrogen bonds and π - π stacking in sorafenib-IR783 nanoaggregates. Transmission electron microscopy, X-ray scattering, and energy-dispersive X-ray spectroscopy coupled with scanning transmission electron microscopy suggested an amorphous core-shell structure for the nanoaggregates.

Overall, the authors present a novel approach to understand the structural features of molecules that drive nanoaggregation. This work adds a mechanistic understanding to previous studies on predicting drug nanoaggregation using correlation with molecular descriptors and other computational approaches. It is recommended that this manuscript is accepted for publication after the following minor revisions:

1. Abstract: The first sentence of the abstract should be revised for grammar.
2. Line 93: The phrase "diameters of 500 nm" should be "diameters smaller than 500 nm".
3. Figure 1: In the caption and the text describing data in this figure, it isn't immediately clear which excipient indocyanine dye is used in each experiment (eg. was a dye-stabilized formulation used for AFM?). This should be clearly labelled on figure panels and/or captions and in the manuscript text.
4. Figure 1b: The "control" used here should be specified.
5. Figure 1d: In the caption, the fragment number of biphenyl-4-carboxylic acid should be included for consistency.

6. Line 144: It should be specified which fragments from Supplementary Table 1 are being referred to that form micron size precipitates.

7. Line 193-198: The authors could hypothesize why ortho-substitution resulted in soluble drug whereas meta-substitution resulted in precipitation when calculated hydrophobicity was identical.

8. Figure 2a,b: The dye used to stabilize these fragments should be mentioned in the figure caption.

9. Line 253: The figure reference here could be more clear since fragment 28 isn't present in Figure 1e and it's unclear what part of Figure 3 is being referred to.

10. Figure 4f: Using colours that are more distinct would help to differentiate the "Sorafenib" and "Sorafenib-IR783 (Lyophilized)" groups.

11. Line 348: The calculated radius of gyration should be discussed in relation to the size measured by DLS.

12. Figure 5: The non-spherical shape of regorafenib aggregates should be discussed.

13. Discussion: The authors should elaborate on how increased dissolution rates in these amorphous structures could impact the use of nanoaggregates in a delivery approach, since nanoparticle dissolution would impact the pharmacokinetics and/or biodistribution of the drug.

Reviewer #2 (Remarks to the Author):

In this manuscript, the authors use a variety of experimental methods to investigate the interactions which drive the self-assembly of small molecules. In doing so they identify two key classes of interactions, pi-pi stacking and hydrogen bonding. Of course, this is not particularly surprising as these interactions stabilise/drive the self assembly of all sorts of soft matter/biological matter including surfactants, polymers and proteins. So while interesting I don't find that to be a surprising result. In fact there are some recent studies in which these exact interactions led to the aggregation of drugs within

micelles (Li & Yang, *Micro and Nano Carriers* (2015) 32, 255-272; Ishkhanyan et al, *J Molec. Liquids* (2022) 356,119050 & Ishkhanyan et al. *Nanoscale* (2022) 14, 5392-5403) so while such a systematic investigation has not been conducted these driving forces have already been identified for the aggregation of small molecule therapeutics.

The experimental investigation of sorafenib and its analogues is a nice approach to demonstrate some of the knowledge obtained from their investigation of the fragments. However, I don't think that the molecular dynamics simulations are particularly insightful. The simulations are only 100 ns in length there is no evidence that this results in an aggregate that is converged on a consistent structure or if it is still a transient object. Also the analysis of the systems that have been done is pretty standard and doesn't really add anything new to the story as it is. I think it should just be removed.

In the conclusions the authors suggest that drug nanoaggregates form a structure that is representative of an amorphous core-shell structure. In the references above, there are also suggestions of such structures for a few other systems. And with the work investigating one system here, it is definitely something to continue investigating but I don't think that the work presented here is enough to result in a conclusive understanding that will always be the case.

In short, I think the experimental investigation of the fragments and the ability to use it to carry out an informed investigation of a specific drug and its analogues to investigate the driving forces which result in the self-assembly of small molecule therapeutics is interesting. I think that in their current state the simulations really don't add anything particularly interesting, so I would suggest they can just be removed from the manuscript. And I think the authors should do a more thorough review of the literature as there are examples of studies which have already shown evidence of this general idea, while not being such a systematic study.

Reviewer #3 (Remarks to the Author):

In this work, Chen and coworkers identify the structural features of drug molecules that drive successful nanoaggregate formation via an indocyanine dye-based encapsulation strategy that was previously described by the group. The authors identified π - π stacking and hydrogen bonding as key characteristics for nanoaggregate formation, and demonstrate how these principles apply to the self-assembly of a nano-formulation of the drug sorafenib. Interestingly, the authors also show that the sorafenib nanoaggregates possess an amorphous core-shell structure. The work is rigorous and yields useful insights for the rationale design of future drug nano-formulations. However, the manuscript can be further strengthened by addressing the following questions:

1. The analysis in this paper focuses on the molecular characteristics of the drug molecule. What is the role of the indocyanine dye in understanding these trends? Do the authors anticipate that their results would still be true with a different stabilization mechanism?
2. To test for colloidal stability, the aggregates are resuspended in distilled water instead of PBS, which may be more relevant at the point of delivery, and which appears to be the protocol used in the group's previous Nature Materials paper. Would a higher ionic strength medium like 1x PBS potentially change the obtained results?
3. What is the rationale for focusing on IR783 and ICG dyes? What are the differences between the two in terms of nanoaggregate formation efficiency?
4. Based on the subsequent findings in the paper, why do the authors think that Fragments 1 and 2 (Figure 1b and 1e) did not aggregate and precipitated, respectively?
5. Are there differences in the percent yield of nanoaggregates formed from the different substructures? It seems that the authors have only reported a binary metric (did aggregates form or not) for nanoaggregate formation.
6. In extrapolating the results from the fragment study to the drug sorafenib, the authors assume that a diphenyl ether substructure is analogous to a 4-phenoxy pyridine substructure. Why is this a fair assumption? Is it possible to test the fragment 4-phenoxy pyridine alone? If that fragment indeed also forms nanoaggregates, then are there differences (e.g. in aggregate size) compared to the diphenyl ether?

Minor comments/questions:

7. In Figure 1e, there appears to be substantial variation in the size of the nanoaggregates formed by Fragments 3 through 6. What do the authors think accounts for this size variation?
8. I find the arrows in Figure 3 to be confusing because they suggest a process or synthetic flow. Please consider an alternative way to convey the information visually.
9. IR125 is referred to as ICG throughout the body of the manuscript, but as IR125 in the Methods (particularly "Nanoaggregate synthesis and characterizations"). Please be consistent with the naming throughout.
10. Typo: "parallel staked" > "parallel stacked"
11. In Figure 5, the authors show EDS maps of trametinib-ICG and regorafenib-ICG. What are the characteristics (size, colloidal stability) of these other drug aggregates? It would also be helpful to explicitly spell out how these drugs also satisfy the molecular characteristics that the authors have identified as being important for nanoaggregate formation.

Reviewer #1 (Remarks to the Author):

The manuscript by Chen et al. investigates the structural features of molecules that allow formation of nanoaggregates using small molecule fragments and structure-assembly-relationship analysis. The authors first evaluate aromatic small molecule fragments for self-assembly with indocyanine excipients and demonstrate the importance biphenyl structures and the presence of hydrogen bonding and π - π stacking interactions for nanoaggregation. Investigation of structural analogs of a known nanoaggregator, sorafenib, further supported the importance of these structural features through structure-assembly-relationship analysis. Molecular dynamics simulations support the existence of hydrogen bonds and π - π stacking in sorafenib-IR783 nanoaggregates. Transmission electron microscopy, X-ray scattering, and energy-dispersive X-ray spectroscopy coupled with scanning transmission electron microscopy suggested an amorphous core-shell structure for the nanoaggregates.

Overall, the authors present a novel approach to understand the structural features of molecules that drive nanoaggregation. This work adds a mechanistic understanding to previous studies on predicting drug nanoaggregation using correlation with molecular descriptors and other computational approaches. It is recommended that this manuscript is accepted for publication after the following minor revisions:

We thank the reviewer for their thorough reading of the manuscript and for their feedback. We have now addressed these comments in the revised manuscript.

1. **Abstract:** The first sentence of the abstract should be revised for grammar.

We have changed the first sentence for clarity, and it now reads “Drug nanoaggregates are particles that can deleteriously cause false positive results during drug screening efforts, but alternatively, they may be used to improve pharmacokinetics when developed for drug delivery purposes.”

2. **Line 93:** The phrase "diameters of 500 nm" should be "diameters smaller than 500 nm".

We have now changed the phrase as suggested.

3. **Figure 1:** In the caption and the text describing data in this figure, it isn't immediately clear which excipient indocyanine dye is used in each experiment (eg. was a dye-stabilized formulation used for AFM?). This should be clearly labelled on figure panels and/or captions and in the manuscript text.

We have now clarified the figure caption to indicate that IR783 was used for the formulation.

4. **Figure 1b:** The "control" used here should be specified.

We have now specified that “Control” means “no excipient” in the figure caption.

5. **Figure 1d:** In the caption, the fragment number of biphenyl-4-carboxylic acid should be included for consistency.

We have added the fragment number (Fragment 3) in the figure caption.

6. **Line 144:** It should be specified which fragments from Supplementary Table 1 are being referred to that form micron size precipitates.

We have now indicated the fragment number (Fragment 17-19) in the main text.

7. **Line 193-198:** The authors could hypothesize why ortho-substitution resulted in soluble drug whereas meta-substitution resulted in precipitation when calculated hydrophobicity was identical.

We speculate that there are two potential reasons that can result in the discrepancies between observed solubility and calculated hydrophobicity. First, the algorithm for the calculation of hydrophobicity, in this case ClogP, is structurally fragmented and additive. Therefore, the location of the functional groups would not be differentiated, and regioisomers can thus have the same ClogP. However, the location of the functional groups can effectively change the steric effects on the molecules, resulting in different solvent exposure surface areas. These differences can significantly impact solubility. Also, ClogP predicts the partition coefficient of water/octanol mixtures, two immiscible solvents. However, the nanoaggregates initially formed in a 7% DMSO-water, two miscible solvents. Studies have shown that DMSO can alter the molecular conformation of certain hydrophobic small molecules in water (*Ghosh, J. Phys. Chem B, 2011*).

We have now changed added an explanation in the main text pasted below:

(Results) Regioisomers had substantial differences in CAC despite identical calculated hydrophobicity values (ClogP or intrinsic solubility, Supplementary Table 2). **The discrepancies are possibly due to the location of the functional groups, which can effectively change the steric effects, and are further affected by the 7% DMSO in the aggregation formation condition.²⁸ These conformational changes, resulting in different solvent exposure surface areas, cannot be predicted using calculated hydrophobicity values that rely on LogP computation of separated atoms or predefined fragments.^{29,30}**

8. **Figure 2a,b:** The dye used to stabilize these fragments should be mentioned in the figure caption.

We have now clarified the figure caption to indicate that IR783 was used in the formulation.

9. **Line 253:** The figure reference here could be more clear since fragment 28 isn't present in Figure 1e and it's unclear what part of Figure 3 is being referred to.

We thank the reviewer for pointing this out. We rearranged and clarified the numbering of the compounds. We also added additional fragments (e.g. 4-phenoxy pyridine, 4-phenoxy pyridine-2-carboxylic acid, 4-pyridin-4-yloxybenzoic acid) to better present the SAR. We reconstructed Figure 3 as shown below, updated the Supplementary Table 1, and pasted the text (below) that is relevant to these changes.

Sorafenib

27
Solution

28
Solution

29
Solution

30
Precipitate

31
Precipitate

5
Nanoaggregate

32
Nanoaggregate

33
Nanoaggregate

34/35
Precipitate

36
Precipitate

Figure 3. Structure-assembly-relationship of sorafenib nanoaggregate formation. Structures in red are potential core scaffolds for π - π stacking, and functional groups in green are potential hydrogen bonding moieties. DLS data are shown for molecules that formed nanoaggregates with IR783 and are absent for the others. Molecules that precipitated or dissolved into the solution are denoted as “Precipitate” or “Solution”.

(Results) We initially investigated the structural dependence of scaffolds for π - π stacking, and the position of hydrogen bonding moieties, on sorafenib-IR783 nanoaggregate formation. Based on our initial fragment-based studies above, we surmised that 4-phenoxy pyridine (Fragment 27) in sorafenib, which resembles the biphenyl groups in a conjugated scaffold that promotes π - π stacking (Fig. 3). In sorafenib, hydrogen bond-forming functional groups are located at both positions 3 and 4 of the 4-phenoxy pyridine moiety. However, the carboxylic acid-substituted 4-phenoxy pyridine at position 3 (Fragment 28) or at position 4 (Fragment 29) did not form nanoaggregates because they were soluble. Therefore, to increase the hydrophobicity of the core scaffold, we investigated diphenyl ether (Fragment 30) as the π - π stacking scaffold and examined the significance of the hydrogen bond functional group positions. Like the meta-substituted biphenyl carboxylic acid (Fig. 2a, Fragment 20), the carboxylic acid substituted 4-phenoxy pyridine at position 3 (Fig. 3, Fragment 31) did not form nanoaggregates. This result suggests that the N-methyl amide in sorafenib is not critical for hydrogen bond formation. Instead, the carboxylic acid-substituted diphenyl ether at position 4 (Fragment 5) formed nanoaggregates (Fig. 3), associating the location of hydrogen bonding to the assembly of sorafenib nanoaggregates.

10. **Figure 4f:** Using colours that are more distinct would help to differentiate the "Sorafenib" and "Sorafenib-IR783 (Lyophilized)" groups.

We have now changed the color of “Sorafenib” group to improve the visualization of this graph.

11. **Line 348:** The calculated radius of gyration should be discussed in relation to the size measured by DLS.

We have now added a comparison for the radius of gyration and hydrodynamic radius in the main text:

(Results) The radius of gyration ($R_g=250 \pm 63.4 \text{ \AA}$) of sorafenib-IR783 nanoaggregates is comparable to its hydrodynamic radius, measured by DLS ($35.32 \pm 2.05 \text{ nm}$) (Supplementary Fig. 7).

12. **Figure 5:** The non-spherical shape of regorafenib aggregates should be discussed.

We thank the reviewer for pointing out the difference in the regorafenib-ICG sample. We believe that, due to the high concentration of the samples, and the requirement for dry samples for TEM, all of the TEM grids appeared to have multiple nanoaggregates in the same image frame. Therefore, three regorafenib-ICG nanoaggregates appear in Figure 5, thus appearing non-spherical. We have now discussed this issue in the main text to avoid the confusion:

(Results) Multiple regorafenib-ICG nanoaggregates appear in the HAADF-STEM image, thus resulting in the observed morphology (Fig. 5a).

13. **Discussion:** The authors should elaborate on how increased dissolution rates in these amorphous structures could impact the use of nanoaggregates in a delivery approach, since nanoparticle dissolution would impact the pharmacokinetics and/or biodistribution of the drug.

We thank the reviewer for this suggestion and have now expanded the discussion to elaborate on the utility of nanoparticle dissolution, with appropriate citations.

Drugs with poor solubility, particularly those classified as Class II or IV pharmaceuticals, require extensive formulation development to enhance their absorption, facilitate passage across biological barriers, and maintain efficacious drug concentrations in the body.⁵¹ Nanoaggregates with amorphous structures, which can offer large surface areas, can thereby raise the dissolution rates of these drugs while preventing undesired precipitation in biological fluids.^{19,51,52} Previous research has demonstrated that colloidal nanoaggregates exhibit superior stability in serum compared to their free drug counterparts, and the use of amorphous dispersions effectively enhances plasma drug exposure.^{53,54}

Reviewer #2 (Remarks to the Author):

In this manuscript, the authors use a variety of experimental methods to investigate the interactions which drive the self-assembly of small molecules. In doing so they identify two key classes of interactions, pi-pi stacking and hydrogen bonding. Of course, this is not particularly surprising as these interactions stabilise/drive the self assembly of all sorts of soft matter/biological matter including surfactants, polymers and proteins. So while interesting I don't find that to be a surprising result. In fact there are some recent studies in which these exact interactions led to the aggregation of drugs within micelles (Li & Yang, *Micro and Nano Carriers* (2015) 32, 255-272; Ishkhanyan et al, *J Molec. Liquids* (2022) 356,119050 & Ishkhanyan et al. *Nanoscale* (2022) 14, 5392-5403) so while such a systematic investigation has not been conducted these driving forces have already been identified for the aggregation of small molecule therapeutics.

We thank the reviewer for their careful critique of this work. We agree that many self-assembled material systems are driven by similar molecular interactions, such as hydrogen bonding or hydrophobic-hydrophobic interactions. However, the novelty here derives from several findings:

First, although a significant amount of work has been done with micelles, a liquid colloid (e.g. micelle, liposome) is structurally and thermodynamically different from a solid colloid (e.g. nanoaggregate). A work previously published at the Shoichet group (Duan, *ACS Chem. Biol*, 2017) demonstrated extensively that a colloidal drug aggregate is distinct from a micelle-like structure. In their study, they used small angle X-ray scattering to investigate the pair distance distribution function of the drug aggregates, revealing the particles are well-packed throughout, where micelles typically have skewed interatomic distance distribution. They also used multi-angle light scattering to compute the R_g/R_h ratio. Interestingly, nanoaggregates, with R_g/R_h ratio less than 1, behave more like hard spheres (e.g. polystyrene beads) than polymeric micelles that have R_g/R_h ratio >1 . In addition, a review on these types of nanoaggregates (Reker, *Nat. Chem.* 2019) also mentioned that the molecular frameworks that lead to drug aggregation are "currently not understood." In our previous study (Shamay, *Nat. Mater.*, 2018), we screened a panel of surfactants and polymers (SDS, SDBS, SDC, PSS, etc.) to formulate drug nanoaggregates, and they all failed to form colloids. Therefore, we speculate that there are still mechanistic nuances that are heretofore not understood.

Additionally, previous studies have primarily focused on examining interactions between excipients or excipients-drugs such as surfactants, polymers, and proteins within micelles or other biological soft matters. These molecules have highly specialized physicochemical properties and structures, such as amphipathic properties and hydrophobic tails. In contrast, our study specifically investigates drug-drug interactions. As we mentioned in the introduction, these drug-drug interacting nanoaggregates can form with or without the presence of excipients. Moreover, the chemical space of small molecule leads/hits is considerably more diverse as compared to excipients. The significance of this study extends beyond drug delivery and formulation science. It also holds relevance for drug screening, as small molecule nanoaggregates can spontaneously form in screening well plates even without excipients, leading to false positive hits in many high throughput studies. Our work aims to provide a mechanistic explanation for the occurrence of this phenomenon.

Last, we agree that it is important to draw analogy/differentiation to micelles. We have extended our discussion to address these points:

(Discussion) Similar molecular interactions were found in surfactant micelles, where π - π stacking and hydrogen bonding also play important roles for the self-assembly of liquid colloids.⁴¹⁻⁴⁵ A polar group or a hydrophilic group usually remains hydrated in surfactant micelles,^{42,43} but we discovered that these polar and hydrophilic functional groups can also stabilize the solid colloids via intermolecular dimerization, and potentially phase separated from the liquid.

The experimental investigation of sorafenib and its analogues is a nice approach to demonstrate some of the knowledge obtained from their investigation of the fragments. However, I don't think that the molecular dynamics simulations are particularly insightful. The simulations are only 100 ns in length there is no evidence that this results in an aggregate that is converged on a consistent structure or if it is still a transient object. Also the analysis of the systems that have been done is pretty standard and doesn't really add anything new to the story as it is. I think it should just be removed.

We thank the reviewer for this suggestion. Before removing the computational work completely, we want to elaborate the purpose of the section. While structural-assembly-relationship studies chemically demonstrate the driving forces of nanoaggregate self-assembly, we wanted to provide a quantitative picture of the internal nanoaggregate arrangement. We used All-Atom Molecular Dynamics simulations to visually elucidate the potential interactions in the nanoaggregates. We have now included additional data that supports convergence. We calculated the average molecular distance over simulation time and found that the distance stabilized around 20 ns.

Supplementary Figure 6. Average molecular distance calculated over simulation time, of sorafenib and IR783 molecular interactions. Error bar = SEM.

(Results) The 100 ns simulation consisted of four IR783 molecules and twelve sorafenib molecules in a box with explicit water, and the simulation reached equilibrium at around 20 ns, based on calculation of the average molecular distance (Supplementary Fig. 6).

(Methods) Topologies of IR783 and Sorafenib were generated using the general Amber forcefield. The forcefield was selected as it is suggested for small hydrophobic molecules. Amber topologies were converted to Gromacs format using ParmEd to run gpu-accelerated simulations on Gromacs. All Atom Molecular Dynamics (AAMD) simulations were run with explicit solvent, using the TIP3 model, at neutral conditions. Twelve molecules of sorafenib and four molecules of dye were placed randomly in a five-nanometer box. This ratio was chosen to match experimental molar equivalents of drug and dye prior to mixing. The energy of the system was minimized to ensure that there were no steric clashes. NVT and NPT equilibration was conducted for 100 ps to equilibrate solvent molecules around the dye and drug. The equilibrated system was run for 100 ns and the coordinates were saved every 200 ps, for a total of 500 frames. Customized cpptraj codes, written using C++, were used to measure the average distances of each molecule of IR783 or sorafenib in the simulation.

In the conclusions the authors suggest that drug nanoaggregates form a structure that is representative of an amorphous core-shell structure. In the references above, there are also suggestions of such structures for a few other systems. And with the work investigating one system here, it is definitely something to continue investigating but I don't think that the work presented here is enough to result in a conclusive understanding that will always be the case.

We agree with the reviewer's comment. We have now changed the discussion to include the potential for caveats in other nanoaggregate systems. In addition, we added a more open-ended discussion to encourage future studies to investigate the structures of other nanoaggregate systems:

(Discussion) Previous works suggest that these colloidal nanoaggregates have filled and non-hollow structures, as opposed to polymeric micelles.^{23,50} Our results added to this knowledge, wherein we found that dye-stabilized nanoaggregates exhibit core-shell structures that are also structurally amorphous (no drug or dye crystallinity). However, whether the structure of nanoaggregates is specific to dye-stabilized structures requires further investigations on other excipients.

In short, I think the experimental investigation of the fragments and the ability to use it to carry out an informed investigation of a specific drug and its analogues to investigate the driving forces which result in the self-assembly of small molecule therapeutics is interesting. I think that in their current state the simulations really don't add anything particularly interesting, so I would suggest they can just be removed from the manuscript. And I think the authors should do a more thorough review of the literature as there are examples of studies which have already shown evidence of this general idea, while not being such a systematic study.

We appreciate this careful review and comments. We have now extended our literature review to other types of nanoparticles and incorporated in the discussion:

(Discussion) Similar molecular interactions were found in surfactant micelles, where π - π stacking and hydrogen bonding also play important roles for the self-assembly of liquid colloids.⁴¹⁻⁴⁵ A polar group or a hydrophilic group usually remains hydrated in surfactant micelles,^{42,43} but we discovered that these functional groups can also stabilize solid aggregates via intermolecular dimerization, and they are potentially phase-separated from the liquid. Intermolecular dimerization through hydrogen bonding has also been observed in certain supramolecular dendrimers, where monomeric dendrons can self-assemble into a hexameric rosette through carboxylic acid dimerization.^{46,47} In addition, π - π stacking as a self-assembly mechanism has been shown in peptide-based drug delivery cargos, where aromatic amino acids like phenylalanine or tryptophan can stack to the aromatic moieties in drugs or nucleic acids.^{48,49} Similarly, we found that indocyanines dyes, which include extensive conjugation and aromaticity, can also localize largely to the surface of nanoaggregates, facilitating colloidal stability.

Reviewer #3 (Remarks to the Author):

In this work, Chen and coworkers identify the structural features of drug molecules that drive successful nanoaggregate formation via an indocyanine dye-based encapsulation strategy that was previously described by the group. The authors identified π - π stacking and hydrogen bonding as key characteristics for nanoaggregate formation, and demonstrate how these principles apply to the self-assembly of a nano-formulation of the drug sorafenib. Interestingly, the authors also show that the sorafenib nanoaggregates possess an amorphous core-shell structure. The work is rigorous and yields useful insights for the rationale design of future drug nano-formulations. However, the manuscript can be further strengthened by addressing the following questions:

We thank the reviewer for their careful reading of this manuscript; we have addressed the comments via point-by-point responses below.

1. The analysis in this paper focuses on the molecular characteristics of the drug molecule. What is the role of the indocyanine dye in understanding these trends? Do the authors anticipate that their results would still be true with a different stabilization mechanism?

We believe that these results may apply broadly to other drug nanoaggregates, especially because others have found a propensity for certain drugs to form them using other excipients. For instance, the Schoichet Lab produced protein-drug aggregates that may be similarly structured, and likewise, nanoaggregates were found in a screen by Reker et al.; both labs found a propensity of similar drugs (including sorafenib) to form nanoaggregates with these other excipients. However, many common surfactants are poor nanoaggregate-formers. In a previous work from our group (Shamay, et. al., *Nat Mater*, 2018), we screened a panel of surfactant and dye excipients, and we found only indocyanine dyes formed nanoaggregates with very high drug loading. We think the large degree of conjugation, large molecular size, and anionic sulfate groups, facilitate stable nanoaggregate formation with indocyanine dyes. We modified our discussion to clarify these points and include additional literature citations:

(Introduction) We previously described a colloiddally stable nanoaggregate platform using an indocyanine dye to encapsulate a wide range of small molecule drugs with unusually high drug loadings. **An indocyanine dye is an amphiphathic excipient that stabilizes drug nanoaggregates.**

(Discussion) **Similar molecular interactions were found in surfactant micelles, where π - π stacking and hydrogen bonding also play important roles for the self-assembly of liquid colloids.⁴¹⁻⁴⁵ A polar group or a hydrophilic group usually remains hydrated in surfactant micelles,^{42,43} but we discovered that these functional groups can also stabilize solid aggregates via intermolecular dimerization, and they are potentially phase-separated from the liquid. Intermolecular dimerization through hydrogen bonding has also been observed in certain supramolecular dendrimers, where monomeric dendrons can self-assemble into a hexameric rosette through carboxylic acid dimerization.^{46,47} In addition, π - π stacking as a self-assembly mechanism has been shown in peptide-based drug delivery cargos, where aromatic amino acids like phenylalanine or tryptophan can stack to the aromatic moieties in drugs or nucleic acids.^{48,49} Similarly, we found that indocyanines dyes, which include extensive conjugation and aromaticity, can also localize largely to the surface of nanoaggregates, facilitating colloidal stability.**

2. To test for colloidal stability, the aggregates are resuspended in distilled water instead of PBS, which may be more relevant at the point of delivery, and which appears to be the protocol used in the group's previous Nature Materials paper. Would a higher ionic strength medium like 1x PBS potentially change the obtained results?

We thank the reviewer for pointing out the differences in the screening conditions. To address this issue, we conducted a colloidal stability test by titrating pH and salt concentrations (for ionic strength). In this experiment, we used the normalized turbidity, an absorbance measurement to survey the concentration of the colloids in the suspension. We discovered that sorafenib nanoaggregates exhibited stability in a range of pHs from 5.5 to 10 and salt concentrations under 5%, which have a greater range than physiological concentrations. However, the exact range for pH and salt concentration also varies based on the drug properties, and many of the fragment-based nanoaggregates are not as stable as drug nanoaggregates, where the formulation has been previously optimized. Therefore, we chose to use water in the screens to account for the idiosyncratic properties of diverse fragments.

We have now included in the pH and salt concentration titration as the Supplementary Figure 4 and described in the Method section and pasted below.

(Results) We chose sorafenib, an FDA-approved kinase inhibitor, to determine how certain functional groups contribute to the intrinsic nanoaggregate formation with the IR783 dye. **Sorafenib-IR783 nanoaggregates exhibit a hydrodynamic diameter of 70.65 ± 4.10 nm and were colloidally stable in water for 7 days (Supplementary Fig. 4), as well as in the pH range of 5.50 – 10.00 and salt concentrations under 5% NaCl (Fig. 3, Supplementary Fig. 4).**

Supplementary Figure 4. Turbidity assessments of sorafenib-IR783 nanoaggregates in (a) a range of buffer pH values and (b) in a range of salt concentrations in water (w/v). Turbidity was normalized to pH=7.4 in panel a and was normalized to water in panel b. N=3 biological replicates.

(Methods) Normalized turbidity measurement sorafenib-IR783 nanoaggregates were aliquoted and redispersed in a range of pH buffer conditions, and a range of salt concentrations using NaCl. Turbidity was measured using absorbance at 600 nm and normalized to each buffer condition without nanoaggregates. Normalized turbidity was calculated using turbidity in pH =7.4 or water with 0% NaCl as a standard, respectively. N=3 biological replicates were performed.

3. What is the rationale for focusing on IR783 and ICG dyes? What are the differences between the two in terms of nanoaggregate formation efficiency?

The rationale for using IR783 is based on our previous research indicating its potential for encapsulating small molecule drugs, and predictability based on the molecular descriptors. However, there were two reasons for exploring ICG dye alongside IR783. ICG is structurally similar to IR783, but not identical. ICG has a larger conjugation system with two additional aromatic rings compared to IR783. By investigating the ICG nanoaggregates, we aimed to understand whether the structural differences between the dyes could influence the self-assembly mechanism. Overall, the nanoaggregate-forming chemical space for ICG is comparable to IR783. However, we observed some exceptions like hydroxyl group. We did not see any trend between IR783 and ICG in terms of encapsulation efficiency for the same fragment/drug. In addition, ICG nanoaggregates have potential clinical implications. ICG is an FDA-approved molecule (though primarily used for medical imaging purposes). We aim to assess potential clinical applications for nanoaggregate drug delivery systems.

We have now included new discussion in the main text and formation efficiency/percent yield in Supplementary Figure 2:

(Results) A similar trend in the size distribution, encapsulation efficiency and loading was observed with nanoaggregates synthesized using ICG instead of the IR783 dye (Supplementary Fig. 1b and Supplementary Fig. 2). However, the hydroxyl substitution was more tolerated in ICG, and we observed formation of smaller nanoaggregates. This result is likely due to more extended conjugation at the backbone of the ICG compared to IR783, such that a less stable intermolecular hydrogen bonding can be compromised through hydrophobic interactions.”

Supplementary Figure 2. Nanoaggregate encapsulation efficiency and fragment/drug loading percent. A selection of fragments and drugs that formed nanoaggregates with IR783 and ICG, characterized for nanoaggregate encapsulation efficiency (a) or fragment/drug loading percent (b). Nanoaggregate encapsulation efficiency (w/w, %) represents the mass percent of fragments or drugs that ended up in the nanoaggregates; fragment/drug loading percent (w/w, %) denotes the mass percentage of fragments or drugs in nanoaggregates. N = 2 biological replicates.

(Results) We previously described a colloiddally stable nanoaggregate platform using an indocyanine dye (IR783) to encapsulate a wide range of small molecule drugs with unusually high drug loadings. Indocyanine dyes are amphipathic excipient that stabilize drug nanoaggregates.¹⁷

(Results) We formulated these fragments with two different indocyanine dyes - IR783 or indocyanine green (ICG); we included the latter because it is, an FDA-approved molecule in the clinic. We analyzed the degree of pelleting after centrifugation, indicating that a nanoaggregate may have formed (Supplementary Fig. 1a).

4. Based on the subsequent findings in the paper, why do the authors think that Fragments 1 and 2 (Figure 1b and 1e) did not aggregate and precipitated, respectively?

We thank the reviewer to point out this question. We speculate that Fragment 1 and 2 failed to form nanoaggregates due to two different reasons. For Fragment 1, even though it contains an aromatic group and a hydrogen bonding moiety, the molecule is too soluble. In Supplementary Table 2, the determined critical aggregation concentration in 7% DMSO for Fragment 1 is larger than 1400 ug/mL. The solubility is unexpectedly high because the DMSO confers much higher solubility to these fragments in our nanoaggregate synthesis condition.

As for Fragment 2, we speculate that naphthalene, like other compounds with fused rings, can π - π stack extremely well due to the rigid backbone and flat surface. The driving forces to form a sheet-like structure are largely preferred, and theoretically the sheet-like structure can expand permitted by the concentration. Therefore, we were seeing large precipitations for Fragment 2. In addition, the presence of rotatable bonds in the encapsulated chemicals may also explain the amorphous nature of the nanoaggregates. For instance, two methyl substitutions of biphenyl (Fragment 25) lock the planes, this additional rigidity can cause the precipitation, similar to Fragment 2. We have now added the explanations in the main text:

(Results) We observed three distinct results: a phenyl or mono-aromatic fragment formed a clear solution with no visible pellets, **as expected from the high CAC of Fragment 1 (Supplementary Table 2)**. A naphthyl or fused-aromatic fragment (Fragment 2) formed visible pellets, but the system was not colloidally stable and settled quickly after resuspension. **We speculate that the instability of fragment 2 is due to its rigid backbone and flat surface that result in its precipitation.**

5. Are there differences in the percent yield of nanoaggregates formed from the different substructures? It seems that the authors have only reported a binary metric (did aggregates form or not) for nanoaggregate formation.

We agree with the reviewer that the assessment of encapsulation efficiency and fragment/drug loading percent of the nanoaggregates should be included. Therefore, we have now included Supplementary Figure 2 (as shown under comment 3).

6. In extrapolating the results from the fragment study to the drug sorafenib, the authors assume that a diphenyl ether substructure is analogous to a 4-phenoxy pyridine substructure. Why is this a fair assumption? Is it possible to test the fragment 4-phenoxy pyridine alone? If that fragment indeed also forms nanoaggregates, then are there differences (e.g. in aggregate size) compared to the diphenyl ether?

We thank the reviewer for correcting our assumption, acknowledging that it was inadequately explained. During the fragment screening process, we utilized 4-phenoxy pyridine as a scaffold, along with its derivatives, such as 4-phenoxy pyridine, 4-phenoxy pyridine-2-carboxylic acid, 4-pyridin-4-yloxybenzoic acid. These fragments exhibited high water solubility due to the presence of an additional nitrogen atom, in contrast to the diphenyl ether derivative fragments. Consequently, no nanoaggregates/precipitations were formed. However, we think it's crucial to compare these primitive fragments to demonstrate the significance of the position of the carboxylic acid. Therefore, we used diphenyl ether instead of 4-phenoxy pyridine as the starting substructure for SAR analysis.

We acknowledge the need for further clarification regarding our decision to utilize diphenyl ether as a base fragment. In response, we have provided a more comprehensive explanation for this choice. We also have incorporated both sets of fragments into Figure 3, and explicitly outlined the rationale behind employing diphenyl ether substructures. As per the reviewer's suggestion in Minor comment 8, below, we have revised Figures 3 to better present the SAR study.

Figure 3. Structure-assembly-relationship of sorafenib nanoaggregate formation. Structures in red are core scaffolds for π - π stacking, and functional groups in green are hydrogen bonding moieties. DLS data are shown for molecules that formed nanoaggregates with IR783 and are absent for those that do not. Molecules that precipitated or dissolved into the solution are denoted as “Precipitate” or “Solution”.

(Results) We initially investigated the potential core scaffolds for π - π stacking and the position of hydrogen bonding moieties on Sorafenib-IR783 nanoaggregate formation. Based on our initial

fragment-based studies above, we surmised that 4-phenoxy pyridine (Fragment 27) in sorafenib, which resembles the biphenyl, functions as a conjugated scaffold that promotes π - π stacking (Fig. 3). In sorafenib, hydrogen bond-forming functional groups are located at both positions 3 and 4 of the 4-phenoxy pyridine moiety. However, the carboxylic acid substituted 4-phenoxy pyridine at position 3 (Fragment 28) or at position 4 (Fragment 29) did not form nanoaggregates due to the hydrophilicity of the fragments. Therefore, with increased the hydrophobicity, we investigated diphenyl ether (Fragment 30) as the core scaffold and examined the significance of the hydrogen bond functional group positions. Like the meta-substituted biphenyl carboxylic acid (Fig. 2a, Fragment 20), the carboxylic acid substituted 4-phenoxy pyridine at position 3 (Fig. 3, Fragment 31) did not form nanoaggregates. This result suggests that the N-methyl amide in sorafenib is not critical for hydrogen bond formation. Instead, the carboxylic acid-substituted diphenyl ether at position 4 (Fragment 5) formed nanoaggregates (Fig. 3), indicating the likely location of hydrogen bonding that enables the assembly of sorafenib nanoaggregates.

Minor comments/questions:

7. In Figure 1e, there appears to be substantial variation in the size of the nanoaggregates formed by Fragments 3 through 6. What do the authors think accounts for this size variation?

We thank the reviewer for this important observation. We found the size of the nanoaggregates can be varied depending on the drug and dye. In our previous work (Shamay, et. al., *Nat Mater*, 2018), we found that the size of the nanoaggregates could be predicted using a molecular descriptor applied to the drug molecule structure. (GETAWAY R4e, which correlates with DLS data with $R^2 = 0.84$, 95% CI [0.22, 0.98]). This descriptor calculates the leverage matrix of a molecule weighted by the intrinsic state. Since the most electronegative groups are often hydrogen bonding, we surmise that there is some relationship between particle size and stability conferred by these groups, but the relationship is not entirely clear. More studies are warranted to dissect these issues, but we decided that doing so within the context of this manuscript would be unwieldy. We agree that it should be noted, however. Thus, we modified the manuscript text to include the following statement:

(Discussion) **We note that the fragments produced nanoaggregates of varying hydrodynamic diameters. We surmise that the size relates to nanoaggregate stability conferred by the drug and dye structures. We note that our previous work found that particle size correlated with some accuracy ($R^2 = 0.84$, 95% CI [0.22, 0.98]) to a molecular descriptor that included an electronegativity term,¹⁷ suggesting some relationship with certain functional groups like hydrogen bonding moieties, potentially analogous to the relationship of particle stability conferred by these groups."**

8. I find the arrows in Figure 3 to be confusing because they suggest a process or synthetic flow. Please consider an alternative way to convey the information visually.

We have now changed Figure 3 to avoid the confusion, as shown under comment 6.

9. IR125 is referred to as ICG throughout the body of the manuscript, but as IR125 in the Methods (particularly "Nanoaggregate synthesis and characterizations"). Please be consistent with the naming throughout.

We have now corrected this in the revised manuscript.

10. Typo: “parallel staked” > “parallel stacked”

We thank the reviewer for identifying the typo, and we have now corrected it.

11. In Figure 5, the authors show EDS maps of trametinib-ICG and regorafenib-ICG. What are the characteristics (size, colloidal stability) of these other drug aggregates? It would also be helpful to explicitly spell out how these drugs also satisfy the molecular characteristics that the authors have identified as being important for nanoaggregate formation.

We have now conducted additional characterization of sorafenib-IR783, trametinib-ICG, and regorafenib-ICG nanoaggregates for size measurement, stability (Supplementary Fig. 10), in addition to the encapsulation efficiency that was shown in Supplementary Figure 2 (Re. Comment 3 or 5). We also modified the text in the revised manuscript (pasted below):

Supplementary Figure 10. Stability of Drug-Encapsulated Nanoaggregates. a-c., The size distribution of the sorafenib-IR783, trametinib-ICG and regorafenib-ICG nanoaggregates by DLS. d-f. Average diameters and polydispersity index (PDI) of the nanoaggregates shown in a-c respectively. N=3.

(Main) The same experiment was also performed on trametinib-ICG and regorafenib-ICG nanoaggregates **that also fit the size and stability criteria set to denote nanoaggregate formation (hydrodynamic diameters of 46.84 ± 12.18 nm and 76.07 ± 3.72 nm respectively, and colloidal stability for at least 3 days, Supplementary Fig. 10)**. Trametinib, regorafenib, and ICG were identified by iodine, chlorine and sulfur atoms, respectively (Fig. 5).

REVIEWER COMMENTS

Reviewer #1 (Remarks to the Author):

The authors addressed all reviewer comments in the revised manuscript. It is recommended that this manuscript is accepted for publication.

Reviewer #2 (Remarks to the Author):

The authors have addressed a majority of my comments thoroughly. The one point I would like to see a bit more care when considering is the equilibration of their simulated systems. In the self assembly of soft matter, there are commonly different equilibration times for different properties of aggregates. In this case I agree that the larger scale properties of the aggregates seem to equilibrate quickly (e.g. the average distance and the SASA), but I am not convinced the internal structure of the aggregates, which the authors have stated is their primary interest in getting from the simulations. In Fig. 4c, the number of pi-pi interactions don't seem to have equilibrated in the simulations as of yet, which one would expect to see some kind of convergence. So I think that the simulations should be extended in order to provide the information with great significance.

More minor suggestions:

In Figures 4c and d, I think these quantities should be plotted as the average number of pi-pi bonds per molecule or the fraction of molecules in pi-pi interactions so that the results are translatable to any system size and not dependent on the number of molecules being studied as of course in the simulations it is a small number of molecules. This will make the values more meaningful to the larger community.

Reviewer #3 (Remarks to the Author):

The authors have addressed my concerns.

A point-by-point response to the reviewers' comments

Reviewer #1 (Remarks to the Author):

The authors addressed all reviewer comments in the revised manuscript. It is recommended that this manuscript is accepted for publication.

We appreciate the reviewer's assistance in facilitating improvements to the manuscript.

Reviewer #2 (Remarks to the Author):

The authors have addressed a majority of my comments thoroughly. The one point I would like to see a bit more care when considering is the equilibration of their simulated systems. In the self assembly of soft matter, there are commonly different equilibration times for different properties of aggregates. In this case I agree that the larger scale properties of the aggregates seem to equilibrate quickly (e.g. the average distance and the SASA), but I am not convinced the internal structure of the aggregates, which the authors have stated is their primary interest in getting from the simulations. In Fig. 4c, the number of pi-pi interactions don't seem to have equilibrated in the simulations as of yet, which one would expect to see some kind of convergence. So I think that the simulations should be extended in order to provide the information with great significance.

We thank the reviewer for their constructive assessment of the manuscript and our responses, and we agree with the reviewer that the simulation timeframe requires further investigation. We extended the simulation to 200 ns and introduced new analyses to assess convergence of the internal structure of the nanoaggregates. To assess the equilibration of the nanoaggregates, we computed a kymograph of all hydrogen bonding molecular interactions in the simulation (**Supplementary Figure 7a**). Urea-urea intermolecular H-bonding was the only type of interaction that exhibited a prolonged residence time, with the first appearance around 20 ns. To investigate the degree of internal equilibrium of the nanoaggregates via the simulation, we extrapolated the kymograph and fitted it as an exponential plateau function. We found that the function asymptotically approached a maximum number of hydrogen bonding interactions of 195.4, 95% CI [189.8, 199.5]. The exponential plateau function followed a similar trajectory when fitted using the original 100 ns simulation (**Reviewer-only Figure**), where the function asymptotically approached 191.5, 95% CI [177.0, 211.0] as the maximum number of H-bonding interactions. Hence, we conclude that the exponential plateau function is a robust model to predict the maximum number of H-bonding interactions.

Based on the kymograph and the function trajectory, we conclude that the reviewer correctly identified that 100 ns simulation was too short to reach an internal equilibrium, at least via the calculation of hydrogen bonding interactions. In the original 100 ns simulation, the observed maximum number of H-bonding interactions was 157 at the 100 ns mark (only 82% of total H-bonding interactions as per the asymptote of 100 ns simulation). On extending the simulation to 200 ns, the maximum number of observed H-bonding interactions rose to 186 (96% of total H-bonding interactions predicted by the asymptote of 200 ns simulation).

By extending the exponential plateau function from 200 ns to 500 ns, the estimated maximum number of H-bonding interactions increased from 186 to 194 (only a 3% increase of total H-bonding interactions), suggesting the nearly all H-bonding interactions stabilized by 200 ns. Therefore, we conclude that the simulations reached equilibrium by 200 ns. We have included the kymograph as **Supplementary Fig. 7a**, exponential plateau fitting as **Supplementary Fig. 7b**, a **reviewer-only figure** for exponential plateau fitting using 100 ns simulation (as the 100 ns simulation and analysis is now removed from the revised manuscript), and changes to the manuscript text (also pasted below).

We also calculated π - π interactions in the extended simulation. We found that the average number of π - π interactions within the nanoaggregates at 100 ns and 200 ns changed little. By calculating the absolute number of π - π interactions as a percentage of the total number of molecules (16 molecules = 4 IR783 molecules + 12 sorafenib molecules) in π - π interactions (as the reviewer suggested in their comment), we observed a steady oscillation at 77% ($\pm 17.9\%$). As a comparison, the 100 ns simulation reached 73% ($\pm 23.4\%$). We conclude that, in the case of π - π interactions, the additional simulation time improved convergence, but only marginally. Overall, we observed that π - π interactions were much more transient than hydrogen bonding interactions in the simulation, potentially because, compared to hydrogen bonding interactions (2-40 kcal/mol; *Frey, 2004*), π - π interactions are much weaker (0.5 to 2.0 kcal/mol; *Liu, 2018*) and more difficult to conclusively assess in the simulation. We have now included the calculation of π - π interactions and analysis of the 200 ns simulation in **Fig. 4c** and **Supplementary Fig. 7c** (and pasted below) and added discussion accordingly.

Lastly, we updated **Supplementary Figure 6** (Average Molecular Distance) and **Supplementary Figure 11** (Normalized Solvent Accessible Surface Area) to include the 200 ns simulation.

RESULTS:

In order to visualize the hydrogen bonding and π - π interacting motifs in sorafenib-IR783 nanoaggregates, we ran an all-atom molecular dynamics (AAMD) simulation. **The 200 ns simulation** consisted of four IR783 molecules and twelve sorafenib molecules in a box with explicit water. **The simulation reached an initial equilibrium at around 20 ns, based on calculation of the average molecular distances (Supplementary Fig. 6).** In addition, we plotted a kymograph showing the time course of every hydrogen bonding interaction during the 200 ns simulation (Supplementary Fig. 7a). **We identified 186 unique hydrogen bonding at the conclusion of the 200 ns simulation. By fitting the kymograph to an exponential plateau curve, the function reached an asymptote of 195.4 with 95% CI [189.8, 199.5], suggesting that the simulation reached approximately 95% of the maximal number of hydrogen bond interactions within 200 ns (Supplementary Fig. 7b).** Therefore, the results suggest stabilization of the simulated nanoaggregate structure in that timeframe. All four IR783 molecules were located at the surface of the nanoaggregates, and the sorafenib molecules largely localized away from the solvent (Fig. 4a). Although the internal arrangement of sorafenib molecules was largely disordered, we observed clear indications of hydrogen bonding from the urea moiety, and potential π - π interactions (Fig. 4a). We analyzed the interactions of key hydrogen bond-forming functional groups throughout the simulation by calculating the formation of each type of hydrogen bond between sorafenib molecules. We found that intermolecular hydrogen bonds from the urea moieties between two sorafenib molecules occurred at the highest probability as compared to other types (Fig. 4b). **We also quantified likely π - π interactions at every 10 ns during the simulation timeframe, and we measured a centroid distance from 3Å – 5Å to include parallel stacked, parallel displaced, edge-to-face and T-shaped interactions.**^{35,36} We found that, on

average, 77% ($\pm 17.9\%$) of molecules fit these molecular distance and orientation criteria that would permit π - π interactions (Fig. 4c and Supplementary Fig. 8). At each frame, occasional edge-to-face π - π interactions (two or three interactions per frame) were observed, but the majority of the π - π interactions were parallel stacked or parallel displaced. Overall, we observed that π - π interactions tended to be more transient as compared to intermolecular hydrogen bonding in the simulation, possibly owing to the relatively weak nature of π - π interactions.^{37,38} These molecular simulation results support the conclusion that intermolecular hydrogen bonds of the urea functional groups and π - π stackings were dominant interactions in the sorafenib-IR783 nanoaggregates.

Supplementary Fig. 7a. A kymograph of all hydrogen bonding interactions during the course of the MD simulation.

Supplementary Fig. 7b. Non-linear fitting of the kymograph (Supplementary Fig. 7a) to an exponential plateau function. The shaded area indicates the asymptote of 195.4 with 95% CI [189.8, 199.5].

Reviewer-only Figure. Non-linear fitting of the 100 ns simulation kymograph to an exponential plateau function. The shaded area indicates an asymptote of 191.5 with 95% CI [177.0, 211.0].

Figure 4c. Percentage of the total number of molecules involved in π - π interactions during the course of simulation.

Supplementary Fig. 8 Number of π - π intermolecular interactions between sorafenib molecules and between sorafenib and IR783 molecules during the course of the simulation.

Supplementary Figure 6. Average molecular distance calculated at each frame of the molecular dynamics simulation.

Supplementary Figure 11. Normalized solvent accessible surface area of both sorafenib and IR783 during the simulation normalized to time zero.

Methods:

Molecular dynamics simulations and analysis

Topologies of IR783 and sorafenib were generated using the general Amber forcefield. The forcefield was selected as it is recommended for small hydrophobic molecules. Amber topologies were converted to Gromacs format using Parmd to run gpu-accelerated simulations on Gromacs. All-atom molecular dynamics (AAMD) simulations were run with explicit solvent, using the TIP3 model, at neutral conditions. Twelve molecules of sorafenib and four molecules of dye were placed randomly in a five-nanometer box. This ratio was chosen to match experimental molar equivalents of drug and dye prior to mixing. The energy of the system was minimized to ensure that there were no steric clashes. NVT (constant number of atoms, volume, and temperature) and NPT (constant number of atoms, pressure, and temperature) equilibration was conducted for 100 ps to equilibrate solvent molecules around the dye and drug. **The equilibrated system was run for 200 ns and the coordinates were saved every 200 ps, for a total of 1000 frames. Customized CPPTRAJ codes, written using C++, were used to measure the average distances of each molecule and compute the kymograph in the simulation.**

More minor suggestions:

In Figures 4c and d, I think these quantities should be plotted as the average number of pi-pi bonds per molecule or the fraction of molecules in pi-pi interactions so that the results are translatable to any system size and not dependent on the number of molecules being studied as of course in the simulations it is a small number of molecules. This will make the values more meaningful to the larger community.

We thank the reviewer for this suggestion. We have now changed Figure 4d (the new Figure 4c in the revised manuscript) and plotted the y-axis as “percentage of molecules in π - π interactions”. We moved the Figure 4c to Supplementary Fig. 7c to keep the absolute count of the π - π interactions as raw data,

since both figures deliver similar information. Figure 4c and Supplementary Fig. 7c are pasted in the response to Comment 1, above.

Reviewer #3 (Remarks to the Author):

The authors have addressed my concerns.

We appreciate the reviewer's assistance with this manuscript.

REVIEWERS' COMMENTS

Reviewer #2 (Remarks to the Author):

I appreciate the efforts the authors have made to address my concerns and I am now happy to suggest it be published.